# Wnt regulates amino acid transporter *Slc7a5* and so constrains the integrated stress response in mouse embryos

Nadège Poncet[1],[†] (iD), Pamela A Halley[1] (iD), Christopher Lipina[2] (iD), Marek Gierliński[3] (iD), Alwyn Dady[1] (iD), Gail A Singer[1] (iD), Melanie Febrer[4],[‡], Yun-Bo Shi[5] (iD), Terry P Yamaguchi[6] (iD), Peter M Taylor[2] (iD) & Kate G Storey[1],[*] (iD)

## Abstract

Amino acids are essential for cellular metabolism, and it is important to understand how nutrient supply is coordinated with changing energy requirements during embryogenesis. Here, we show that the amino acid transporter *Slc7a5/Lat1* is highly expressed in tissues undergoing morphogenesis and that *Slc7a5*-null mouse embryos have profound neural and limb bud outgrowth defects. *Slc7a5*-null neural tissue exhibited aberrant mTORC1 activity and cell proliferation; transcriptomics, protein phosphorylation and apoptosis analyses further indicated induction of the integrated stress response as a potential cause of observed defects. The pattern of stress response gene expression induced in *Slc7a5*-null embryos was also detected at low level in wild-type embryos and identified stress vulnerability specifically in tissues undergoing morphogenesis. The *Slc7a5*-null phenotype is reminiscent of Wnt pathway mutants, and we show that Wnt/β-catenin loss inhibits *Slc7a5* expression and induces this stress response. Wnt signalling therefore normally supports the metabolic demands of morphogenesis and constrains cellular stress. Moreover, operation in the embryo of the integrated stress response, which is triggered by pathogen-mediated as well as metabolic stress, may provide a mechanistic explanation for a range of developmental defects.

**Keywords** amino acid transport; integrated stress response; mouse embryo morphogenesis; *Slc7a5/Lat1*; Wnt signalling

**Subject Categories** Development; Membrane & Trafficking; Signal Transduction

## Introduction

Embryonic development involves the patterned proliferation of emerging cell populations, with some tissues expanding more rapidly than others to generate new structures, including limb buds and regions of the developing nervous system. Such expansion is driven by the localised activity of key signalling pathways, including fibroblast growth factor and Wnt, e.g. [1–3], and there is growing evidence that links such signalling to the regulation of metabolic gene expression [4,5]. Morphogenesis of the embryo also depends on cell movement and shape changes, and all these cellular processes have high-metabolic requirements that rely on efficient uptake of nutrients including amino acids. Indeed, amino acid supply is essential for survival of pre-implantation mouse embryos [6,7] and both under and over maternal nutrition can retard intrauterine growth and increase the risk of morphogenesis failures, including neural tube defects (reviewed in ref. [8]). Amino acids are provided to cells via membrane-localised transporters, which specialise in the import of specific types of amino acid. However, relatively little is known about the expression pattern, requirement and regulation of such transporters during embryogenesis.

The system L1 amino acid transporter isoforms Slc7a5 and Slc7a8 (aka Lat1 and Lat2, respectively) are the primary (although usually not exclusive) conduits for delivery of large neutral amino acids (LNAAs) to many mammalian cell and tissue types. These include the essential amino acids leucine, isoleucine, phenylalanine, tryptophan, valine and methionine. Slc7a5 and Slc7a8 are Na$^+$-independent amino acid antiporters, typically taking up essential LNAA in exchange for glutamine [9] or histidine [10]. They are also able to transport phenylalanine/tyrosine derivatives such as thyroid hormones $T_3$ and $T_4$ [11] as well as the neurotransmitter precursor L-DOPA [12]. Both Slc7a5 and Slc7a8 genes encode members of the

1  Division of Cell & Developmental Biology, School of Life Sciences, University of Dundee, Dundee, UK
2  Division of Cell Signalling and Immunology, School of Life Sciences, University of Dundee, Dundee, UK
3  Division of Computational Biology, School of Life Sciences, University of Dundee, Dundee, UK
4  Sequencing Facility, School of Life Sciences, University of Dundee, Dundee, UK
5  Section on Molecular Morphogenesis, NICHD, NIH, Bethesda, MD, USA
6  Cancer and Developmental Biology Laboratory, Center for Cancer Research, National Cancer Institute-Frederick, NIH, Frederick, MD, USA
   *Corresponding author. Tel: +44 1382 385691; E-mail: k.g.storey@dundee.ac.uk
   †Present address: Institute of Physiology, University of Zürich, Zürich, Switzerland
   ‡Present address: Illumina Canada, Victoria, BC, Canada

heteromeric amino acid transporter family of proteins which require a regulatory glycoprotein subunit (in this case 4F2hc/CD98/Slc3a2) to function correctly [13,14]. Slc7a5-mediated influx of essential LNAA in particular is recognised to promote net protein synthesis, cell growth and proliferation, reviewed in [15], by processes including activation of the mTORC1 signalling pathway [16,17]. Accordingly, *Slc7a5* is over-expressed in many cancers [18,19] and induction of *Slc7a5* is associated with periods of rapid cell growth and expansion during sustained activation of T lymphocytes [20]. It also plays an important role in maintenance of critical amino acids in the brain [10].

Slc7a5 is clearly implicated in various processes essential for embryonic development such as protein synthesis, cell growth and proliferation, and we have discovered that *Slc7a5*-null mice are embryonic lethal [16]. Early embryos up to and including the blastocyst stage are able to take up LNAA effectively by system B$^{0,+}$ Na$^+$-coupled transport [21]. *Slc7a5* therefore appears to be a good candidate gene for investigating the role and regulation of nutrient and hormone uptake during subsequent embryogenesis and to elucidate how gene regulatory mechanisms influence such environmental factors.

Here, we show that *Slc7a5* expression is patterned in the mouse embryo and that *Slc7a5*-null embryos exhibit profound neural and limb defects. We characterise this phenotype using key tissue and cell type-specific markers and interrogate the *Slc7a5*-null cell state. We detect modest effects on cell proliferation and mTORC1 activity in mutant embryos and uncover an early induction of the integrated stress response (ISR) [22,23]. The ISR is initially adaptive and acts to restore cell homeostasis, but ultimately leads to apoptosis, which is increased in *Slc7a5*-null embryos. Moreover, we show that Wnt signalling is required for *Slc7a5* expression and so prevents ISR induction, supports the elevated metabolic demands of tissue morphogenesis and protects against developmental defects.

# Results

### Slc7a5 is expressed in specific regions of the developing embryo

The spatial and temporal expression pattern of the LNAA transporter *Slc7a5* was assessed by mRNA *in situ* hybridisation in whole mouse embryos from early primitive streak stages (Fig 1); probe specificity was assessed in *Slc7a5*-null embryos, where no signal was detected (Appendix Fig S1). *Slc7a5* mRNA was broadly detected in epiblast, primitive streak and emerging mesendoderm in the embryo at E7.0, as well as in extra-embryonic epiblast and mesoderm [24] (Fig 1A, a1, a2, a2′). At E8.5 (Fig 1B, b1–b6), *Slc7a5* was expressed in the open anterior (Fig 1B, b1, b2), and posterior neural plate, including preneural tube and the caudal lateral epiblast (Fig 1B, b5, b6), and dorsally in closed neural tube (which includes presumptive neural crest) and in somites (Fig 1B, b3, b4). At E9.5, *Slc7a5* transcripts continued in all these domains, with high levels in forebrain and optic vesicle as well as in the otic vesicle and first brachial arch (Fig 1C, D, c1′–c1″), forming cranial ganglia (Fig 1D), dorsal hindbrain and spinal cord (Fig 1c2–c5) and in the progress zone of emerging limb buds (Fig 1E). At E10.5 *Slc7a5* transcripts continued to be detected along the rostro-caudal extent of the developing nervous system at varying levels (Fig EV1), including high

expression in optic and otic vesicles, cranial ganglia (Fig EV1A, a1′–a2′), branchial arches (Fig EV1A, a2) and differentiating somites, neural crest derivatives and mesonephric duct (Fig EV1A, a3). Transcripts were detected more extensively in the limb bud (Fig EV1B, b1–b2). Notably, *Slc7a5* mRNA was most strongly expressed in dorsal spinal cord (Fig EV1a3, a3′) and the forming neural tube arising from the tailbud (Fig EV1C, c1–c5). *Slc7a5* is thus transcribed highly in neural and other tissues that undergo morphogenetic movements and/or proliferative expansion in the developing embryo.

### Slc7a5-null embryos exhibit morphological neural tube and limb defects

To determine the requirement for *Slc7a5* during embryogenesis, null embryos were generated by inter-crossing heterozygote *Slc7a5*$^{+/-}$ mice (see Materials and Methods). An abnormal phenotype was first apparent at E9.5 (Fig 2A–H′). Mutant embryos appeared smaller than littermates and exhibited failure of zippering up along the fore- and midbrain and closure of the neural tube at site 2 (forebrain/midbrain boundary) (Fig 2B, F, F′) as well as a delay of posterior neuropore closure (Fig 2C, E′), while closure at site 1 (at the hindbrain/cervical spinal cord boundary) is completed [25]. In addition, *Slc7a5*-null embryos lacked optic vesicles and had small otic vesicles (Fig 2D, H, H′) and limb buds (Fig 2B and F). In a subset of *Slc7a5*-null embryos, the forebrain formed but failed to expand (Fig EV2). This is similar to the "flat-top" phenotype observed in mTORC1 mutants or rapamycin (mTORC1 complex inhibitor)-treated embryos [26,27]. These overt morphological defects in neurulation and limb bud outgrowth correlate well with the regions in which *Slc7a5* is highly expressed.

### Tissue expansion, neurogenesis and neural crest defects in Slc7a5-null embryos

Marker gene analysis of E9.5 embryos was undertaken next to elucidate these developmental defects. Fibroblast growth factor (*Fgf*) 8 locally regulates expansion of key embryonic tissues, including in the developing brain and limb [28,29] and is co-expressed in the rostral forebrain with *FoxG1/BF1* [30]. These genes were both detected in normal (although reduced) domains in the forebrain in *Slc7a5*-null embryos (Figs 2I, K, and EV3A, B, D, E) and *Fgf8* was similarly present, but in reduced domains at the midbrain–hindbrain boundary and the apical ectodermal ridge which signals to the underlying proliferative progress zone of the limb bud (Fig 2I, J, i1, j1, K, L, k1, l1). As *Fgf8* and *FoxG1* are correctly localised, these data suggest that *Slc7a5* loss does not disrupt overall tissue patterning, but attenuates expansion of cell populations in the developing brain and limb, which can compromise morphogenetic cell movements, such as those underlying neural tube closure [31].

To assess whether neuronal differentiation is affected in *Slc7a5*-null embryos, we analysed expression of the neural progenitor marker *Neurog2* [32] and the neuronal marker *Delta1* [33]. Defective neural tube closure at fore- and midbrain levels made expression patterns difficult to compare with littermate controls, but reduced *Neurog2* expression in the brain and spinal cord were evident (Fig EV3B–F, G–i2), and analysis in the closed neural tube of the spinal cord revealed the lack of dorsally located *Delta1* expressing

cells [34,35] in null embryos (Fig EV3g3, i3, h1, i4). Further, analysis of tubulin-βIII (Tuj-1) expression, which identifies neurons and their extending axons [36], revealed striking failure of axonal outgrowth in the brain and along the length of the neural tube in *Slc7a5*-null embryos (Fig 2M–n1). This was particularly evident in neural crest-derived cranial and spinal ganglia (Fig 2M, m1, N, n1),

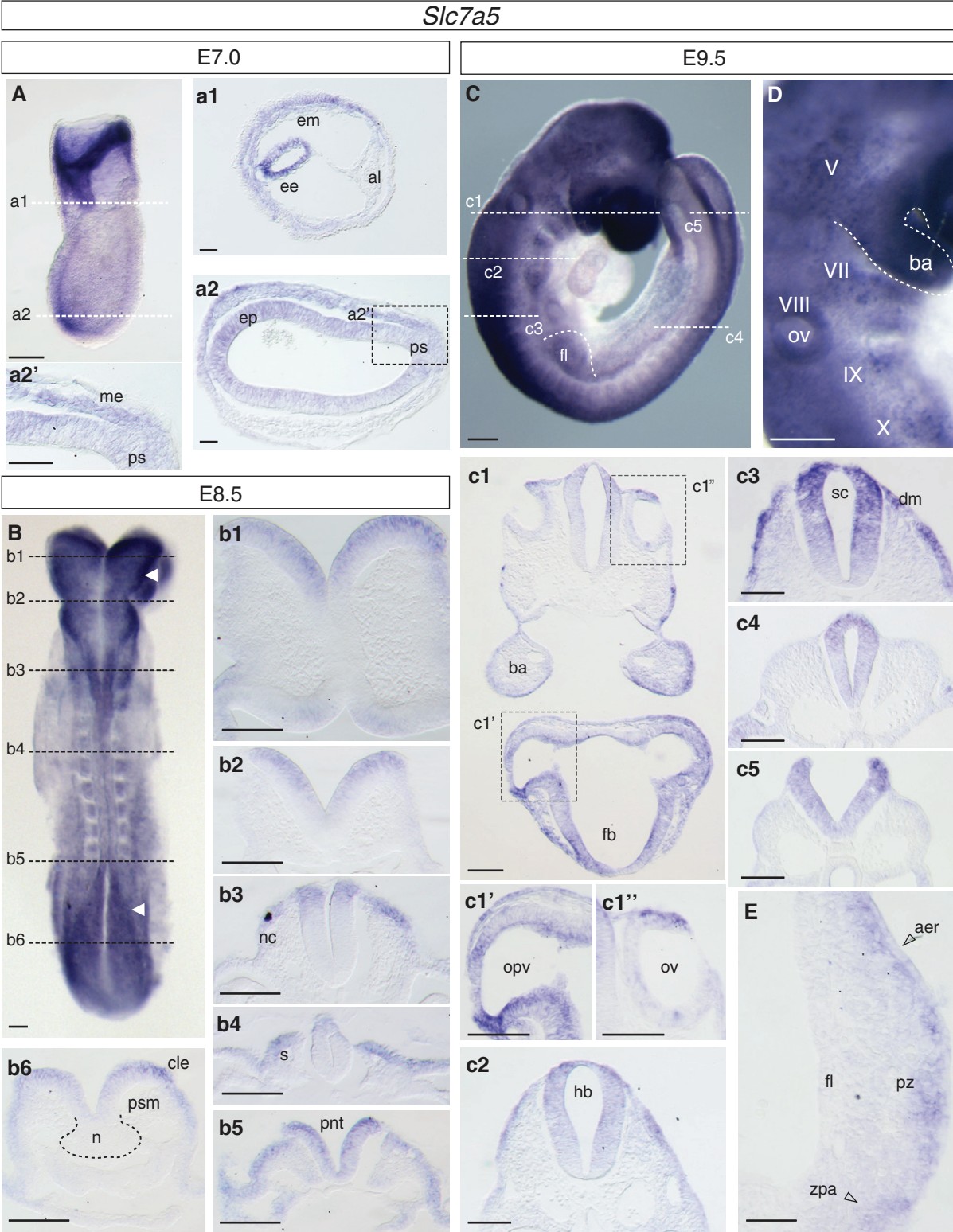

**Figure 1.**

Figure 1. *Slc7a5* mRNA expression pattern in the early mouse embryo.

A E7.0 whole embryo and transverse section (TSs) (dashed lines) through (a1) extra-embryonic tissue showing expression in a ring of extra-embryonic ectoderm (ee) and extra-embryonic mesoderm (em) and absence of transcripts in the allantois (al); (a2) revealing expression in epiblast (ep) and in mesendoderm (me) emerging from the primitive streak (ps), dashed box (a2′) defines higher magnification region shown in a2.

B E8.5 whole embryo dorsal view and TSs (dashed lines) showing expression in: (b1–b3) the open anterior neural plate (future fore-, mid- and hindbrain) including (b3) neural crest (nc) emerging from hindbrain; (b4) in the dorsal neural tube (prospective neural crest) and dorsal somites (s); (b5) in closing preneural tube neural (pnt) and epithelising somites; (b6) in caudal lateral epiblast (cle) adjacent to the node (n) at the anterior tip of the primitive streak, but not in presomitic mesoderm (psm). White arrows indicate open anterior and posterior neural plate.

C E9.5 whole embryo side view and TSs (white dashed lines) showing expression in: (c1) forebrain (fb) and hindbrain, including optic vesicles (opv), (c1′) seen at high magnification (and indicated by box grey dashed line) and otic vesicles (ov) (c1″) seen at high magnification (box grey dashed line); (c2) high expression in dorsal hindbrain (hb) and (c3) spinal cord (sc) and dermamyotome (dm) of the differentiating somite; (c5) throughout open posterior neural plate.

D High magnification of E9.5 whole embryo showing expression in first branchial arch (ba) and forming cranial ganglia, which are just becoming apparent, (V), (VII), (VIII) (IX) and (X). OV, otic vesicle.

E Expression in mesenchyme cells underlying the apical ectodermal ridge (aer) in the progress zone (pz) and in the zone of polarising activity (zpa) of the forelimb (fl).

Data information: Images representative of $n \geq 4$ embryos for each stage. Scale bars 200 μm, except sections 100 μm.

and this was consistent with reduced *Neurog2* expression in these cell populations (Fig 2O–o4, P–p4) [37].

To dissect further the requirement for *Slc7a5* in neural crest, *Sox10* expression was used to identify migrating neural crest and its derivatives [38,39]. *Sox10* expression domains were less extensive in *Slc7a5*-null embryos (Fig 3A and B) including cranial as well as spinal ganglia (Fig 3a4, b4). We next assessed pre-migratory neural crest cells within the dorsal neural tube itself. These cells express Pax3, which acts early in the neural crest gene regulatory network [40,41]. The domain of Pax3 expression in the dorsal neural tube was reduced in *Slc7a5*-null embryos (Fig 3C and D, quantified in 3G), suggesting a role for *Slc7a5* in the self-renewal of the pre-migratory pool of neural crest precursors [42]. In contrast, expression of FoxA2 in the ventral neural tube and notochord appeared unaltered in *Slc7a5*-null embryos (Fig 3E and F). These data identify specific defects in the expansion and differentiation of cells in the dorsal neural tube in *Slc7a5*-null embryos. Overall, this marker gene analysis indicates a requirement for *Slc7a5* in cell populations undergoing energetic activities, differentiation, extensive movement, cell shape change or expansive growth during the morphogenesis phase of embryonic development.

### *Slc7a5*-null embryos exhibit a mild cell proliferation defect and aberrant mTORC1 activity

To assess whether cells in *Slc7a5*-null embryos exhibit reduced proliferation, we next used an antibody against phosphorylated histone 3 (phospho-H3) to identify mitotic cells (Fig 4A–D). We first quantified phospho-H3-positive cells in the spinal cord (at the level of the forelimb) (Fig 4A and B). This revealed no difference in mitotic index in the absence of *Slc7a5* (Fig 4E). The latter region was selected for analysis because mutant embryos complete neurulation in this region and exhibited minimal morphological defects and facilitating comparison with wild-type embryos. We therefore next assessed the mitotic index in the forebrain, where *Slc7a5*-null embryos exhibit profound defects, failing to form optic vesicles and undergo neurulation. Surprisingly, although the neuroepithelium appears depleted in *Slc7a5*-null forebrain, the mitotic index was only just significantly reduced (Fig 4C, D and F). This modest effect in the forebrain and the lack of effect in the spinal cord suggest that reduced cell proliferation is unlikely to be the major explanation for the defects observed in *Slc7a5*-null neural tube.

The resemblance of the forebrain phenotype in a subset of *Slc7a5*-null embryos to the flat-top phenotype resulting from reduced mTORC1 activity [26,27], the requirement for *Slc7a5* for full activation of mTORC1 in some other cellular contexts [16,17,20] along with the well-established role of this pathway in integrating nutrient and mitogen signals [15], raised the possibility that altered mTORC1 signalling might underlie the *Slc7a5*-null phenotype. We took two approaches to assess this possibility. By immunocytochemistry, the phosphorylated form of ribosomal protein S6 (S6 is a P70S6 kinase substrate and often used as surrogate reporter for mTORC1 signalling [43]) was detected (using antibodies recognising phospho-Ser235/236 (Fig 4G and H) or phospho-Ser240/244 (Appendix Fig S2C and D). Strikingly, phospho-S6 levels in neural tube sections varied markedly from cell to cell in *Slc7a5*-null, but not in wild-type embryos (Fig 4G and H, quantified in 4I), indicating a more heterogeneous pattern of mTORC1 activity in the loss-of-function condition. In contrast, Western blot for phospho-Thr389 of P70S6 kinase (a key substrate of mTORC1) did not show a difference between wild-type and *Slc7a5*-null conditions in E9.5 embryo lysates (Fig 4J). This may reflect aberrant rather than absent mTORC1 signalling activity in a subset of cells, which may be beneath detection in whole embryo lysates. Together, these findings suggest that reduced mTORC1 activity is just beginning to be detectable at the time when profound morphological defects are manifest in *Slc7a5*-null embryos.

### Transcriptome analysis implicates the integrated stress response and discounts a role for thyroid hormone in the *Slc7a5*-null phenotype

As we detected only a small effect on cell proliferation in *Slc7a5*-null embryos, we next took an unbiased approach to uncover the earliest effects of *Slc7a5* loss. This additionally allowed us to determine whether *Slc7a5* transport of tyrosine derivatives (including thyroid hormones (TH) $T_3$ and $T_4$) as well as LNAAs might underlie the observed developmental defects. To this end, we carried out RNA-seq on wild-type and *Slc7a5*-null embryos at E8.5, just prior to appearance of phenotypic defects and compared resulting transcriptomes (see Materials and Methods). Significant changes (FDR < 0.05) in the expression of only seven gene transcripts, one pseudogene and two lincRNAs were found in *Slc7a5*-null embryos (Table 1, Datasets EV1 and EV2). These changes were confirmed by

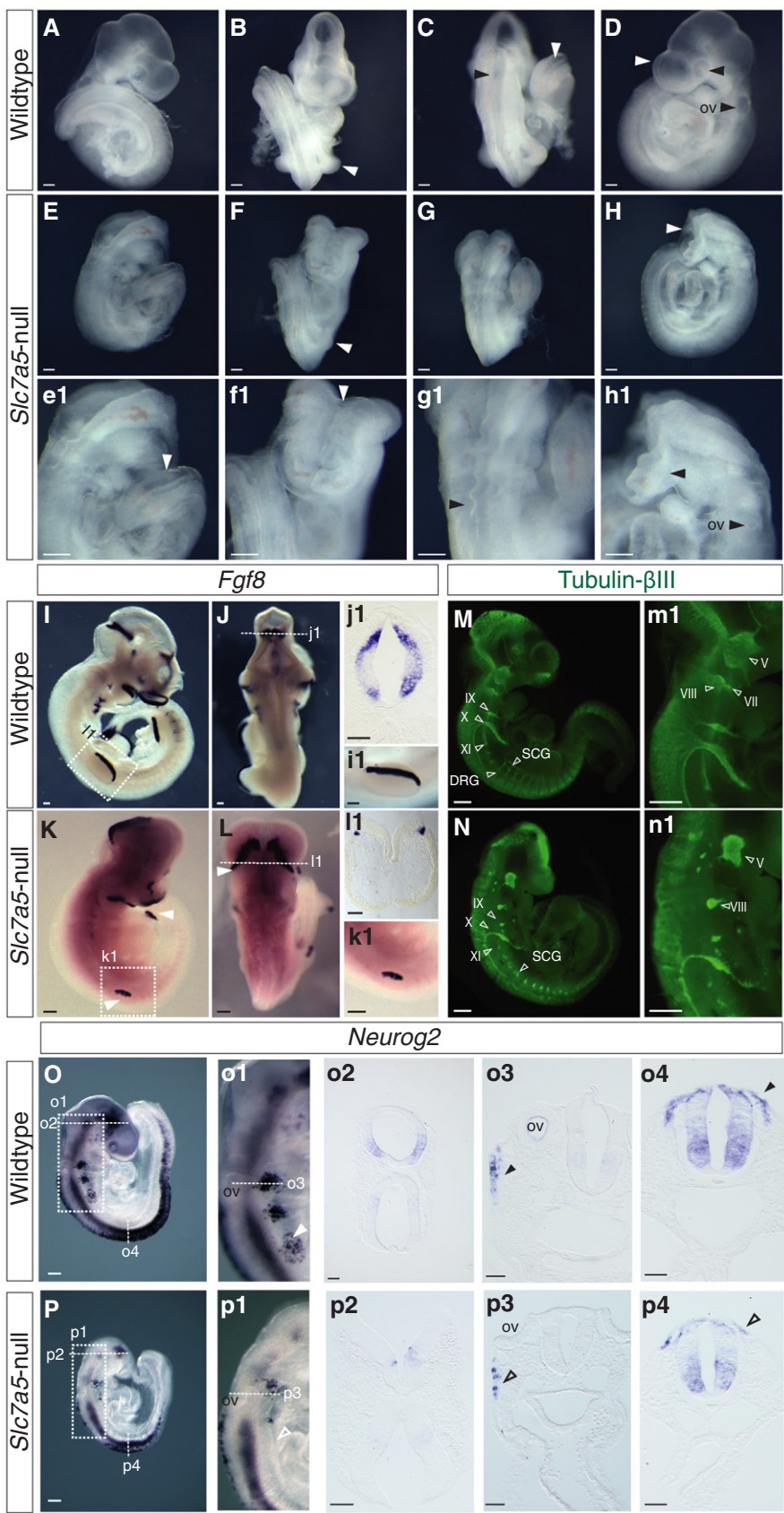

**Figure 2.**

◄

**Figure 2.  *Slc7a5*-null embryos exhibit overt neural tube closure and limb bud outgrowth defects, reduced *Fgf8* expression domains and aberrant neuronal and neural crest differentiation.**

A–H  Live wild-type littermate and *Slc7a5*-null embryos imaged shortly after dissection. (A–D) Wild-type and (E–H) *Slc7a5*-null E9.75 embryos from lateral (A, D, E, H), frontal (B, F) or dorsal (C, G) views; (e1–h1) higher magnification images of the *Slc7a5*-null embryo. White arrowheads indicate the smaller limb bud (compare B and F), open/reduced forebrain (compare D and H), open neural tube at posterior (compare C and E, e1) and anterior (compare B and F, f1) regions. Black arrowheads indicate kinked neural tube (compare C and G, g1) and apparently missing optic and smaller otic vesicles (ov) (compare D and h1).

I–P  mRNA *in situ* hybridisation and immunofluorescence in E9.5 or E10.5 wild-type and *Slc7a5*-null embryos for key marker genes. (I–L) *Fgf8* mRNA transcripts were detected in wild-type (I–j1) and in *Slc7a5*-null (K–l1) E10.5 embryos (*n* = 4 each) (white arrowheads indicate limb buds in K and the isthmus in L) with (j1, l1) sections through isthmus at midbrain/hindbrain border and (i1, k1) lateral views of forelimb buds. (M, N) Wholemount tubulin-β-III (Tuj1) immunofluorescence performed on (M, m1) wild-type and (N, n1) *Slc7a5*-null E9.5–E10.5 embryos (*n* = 5 each condition). Dorsal root ganglia (DRG) and sympathetic chain ganglia (SCG) are indicated with open arrowheads together with cranial ganglia IX, X and XI. Images in (m1, n1) show higher magnification of the cranial ganglia V, VII/VIII (open arrowheads). (O, P) *Neurog2* expression in wild-type (O–o4) and *Slc7a5*-null embryos (P–p4) (*n* = 5 and *n* = 6 each), showing reduced expression in forebrain (o2, p2), cranial ganglia (o1, o3, p1, p3, arrowheads) and spinal cord at level of forelimb (o4, p4, arrowheads indicate position of neural crest).

Data information: Scale bars 200 μm, except sections 100 μm.

RT–qPCR for 6 genes: *Slc7a5*, *Klhdc4* and *Spire2* were downregulated and *Chac1, Trib3* and *Pck2* were upregulated (Fig 4K).

These data confirm efficient knockout of *Slc7a5*, while *Klhdc4* and *Spire2* reduction are unlikely to be functionally significant, given the remaining level of expression in *Slc7a5*-null embryos and the existence of other isoforms (see Table 1 and Fig 4K). Analysis in *Slc7a5*-null embryos further confirmed that transcription and protein levels of Slc7a5 partner *Slc3a2/CD98* are not reduced in this context (Appendix Fig S3). In contrast, two of the confirmed significantly upregulated genes *Chac1* (cation transport regulator homolog-1) and *Trib3* (tribbles pseudokinase-3, aka TRB3, NIPK or SKIP3) are implicated in endoplasmic reticulum (ER) stress and the unfolded protein response (UPR) [45–47]. *Chac1* is distantly related to *Botch,* a negative regulator of Notch signalling [48]; however, we found no evidence for change in Notch signalling in *Slc7a5*-null embryos (Appendix Fig S4). Transcription of the third confirmed upregulated gene, *Pck2* (phosphoenolpyruvate carboxykinase-2), is activated by amino acid deficiency and ER stress and also plays a role in gluconeogenesis, a stress-related metabolic pathway [45,49]. In addition, a further two upregulated genes just below significance, *Aldh1l2* (aldehyde dehydrogenase 1 family, member L2) and *Asns* (asparagine synthetase) (Dataset EV1), are also well-known cellular stress-activated genes [50,51]. Multiple stress stimuli, including amino acid deprivation and ER stress, converge on induction of the integrated stress response (ISR) [22,23]. This is an adaptive response which acts to restore cellular homeostasis by decreasing global protein synthesis while promoting mRNA translation for selected proteins. These include the key mediator of the ISR, activating transcription factor (ATF) 4, and this leads to upregulation of ATF4 transcriptional targets, many of which alter cell metabolism so as to mitigate cellular stress [52]. These findings therefore suggest that promotion of the ISR is an early consequence of *Slc7a5* loss in developing embryos.

We further assessed whether altered transport of phenylalanine/tyrosine derivatives might contribute to the mutant phenotype. Indeed, TH signalling has been implicated in expansion of the cerebral cortex at later developmental stages [53]. No changes in genes associated with L-DOPA were found in the RNA-seq data, and while TH target gene *Dio3* was slightly reduced in an analysis in which one outlier data set was removed (Dataset EV2), this was not confirmed in *Slc7a5*-null embryos assessed by qPCR (Appendix Fig S5A). Furthermore, *Dio3* and thyroid hormone receptors THRβ and THRα were barely detectable until after E10.5 in wild-type embryos

(Appendix Fig S5B, D, E, F). These data strongly suggest that the dopamine and thyroid hormone systems do not operate during the period when the *Slc7a5* phenotype is first manifest. Overall, these findings identify loss of Slc7a5 function as a transporter of LNAA as the likely underlying cause of these early neural developmental and limb defects.

### *Slc7a5*-null embryos exhibit localised integrated stress response

To investigate the activation of the ISR in *Slc7a5*-null embryos, qPCR was carried out for key genes that mediate this response in stage E9.5 embryos (when the phenotype is now apparent). This included *Chac1* and *Trib3* and also *ATF4* as well as C/EBP homologue protein (*CHOP*). CHOP is an effector of a pro-apoptotic response which is triggered if cellular stress persists [52,54]. CHOP acts together with ATF4 to promote transcription of *Chac1* and *Trib3* which, along with other stress-induced pathways, promote apoptosis [46,47]. *Chac1*, *Trib3* and *CHOP* were significantly upregulated in *Slc7a5*-null embryos (Fig 4L). *ATF4* transcription showed a similar trend, but was not significantly increased, perhaps reflecting that ATF4 response to stress is primarily at the level of mRNA translation [55]. In addition, Trib3 may act as a feedback inhibitor of *ATF4* transcription [56].

To localise this stress response within the developing *Slc7a5*-null embryo, the expression patterns of *Chac1* and *Trib3* were next assessed by mRNA *in situ* hybridisation. Strikingly, transcripts were found at high levels in *Slc7a5*-null embryos in regions that normally express *Slc7a5* but were not detected in E9.5 wild-type embryos processed in parallel (Fig 5A–H). This was particularly apparent in the brain, branchial arches and otic vesicles as well as dorsal neural tube and limb buds of mutant embryos (Fig 5B, b1–b6, D, F, f1–f6, H). To substantiate further *Trib3* and *Chac1* as markers of cellular stress in the embryo, we additionally analysed their expression following brief exposure of whole E8.5 wild-type embryos to ER stress inducers, tunicamycin and thapsigargin, which also provoke the ISR [57]. This revealed rapid and localised induction of both *Trib3* and *Chac1,* particularly in the neural tube (Appendix Fig S6).

Interestingly, *Chac1* and *Trib3* transcripts could also eventually be detected at low level in wild-type embryos in a similar pattern to that in *Slc7a5*-null, if the mRNA detection reaction was continued for > 3 days (Fig 5I–M and see Appendix Fig S7). These findings strongly suggest that loss of *Slc7a5* promotes localised ISR in the embryo as it undergoes morphogenesis and, given detection of

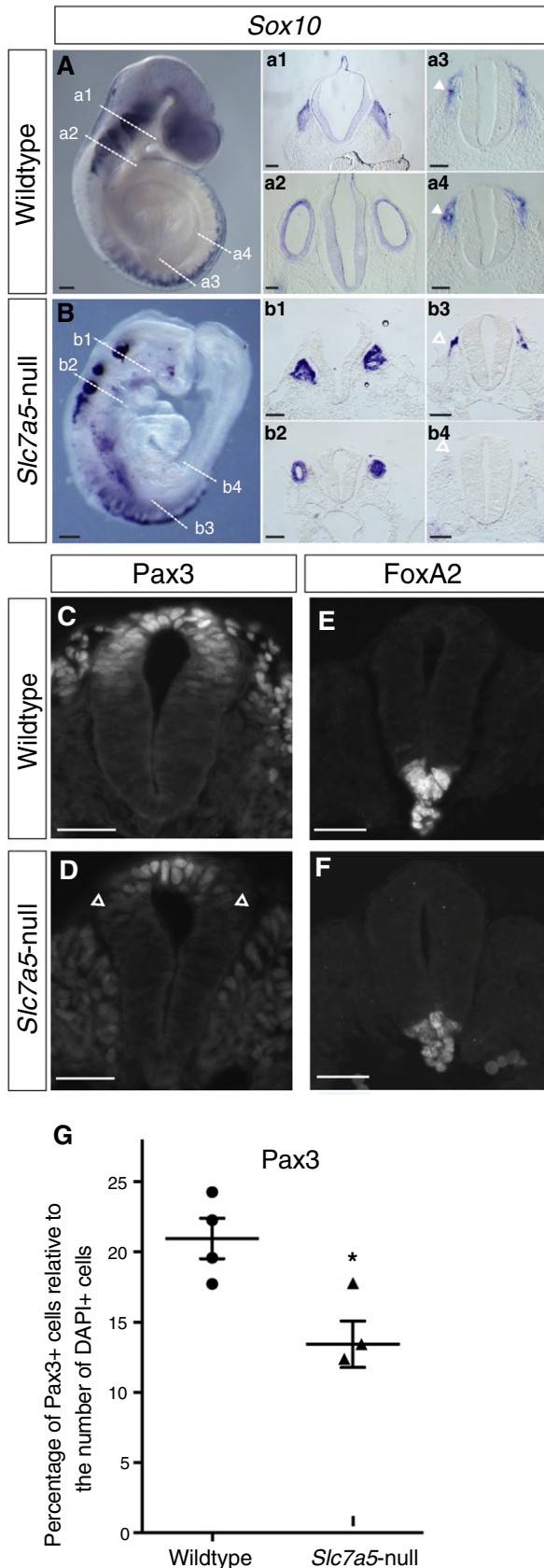

**Figure 3. *Slc7a5*-null embryos exhibit altered expression of neural crest genes.**

A, B   *Sox10 in situ* hybridisation in (A) wild-type and (B) *Slc7a5*-null E9.5 embryos (*n* = 4 each) with TSs of (a1, b1) the hindbrain at level of trigeminal ganglion V, (a2, b2) otic vesicles and the neural tube at more posterior levels (a3, a4, b3, b4). Scale bars 200 μm, except sections a1–b4, 100 μm. Closed white arrowheads in (a3, a4) indicate neural crest, and open white arrowheads indicate depleted neural crest in (b3) and position where neural crest should be in (b4).

C–F   Immunofluorescence on TS caudal spinal cord of E9.5 wild-type littermates and *Slc7a5*-null embryos; Pax3 and FoxA2 were used as indicators for dorso-ventral organisation in (C, E) wild type, *n* = 2 (for FoxA2) and *n* = 4 (For Pax3) and (D, F) *Slc7a5*-null, *n* = 3 (for FoxA2) and *n* = 3 for Pax3) neural tube. Arrowheads indicate border of Pax3 expression domain. Scale bars 50 μm.

G   Percentage of Pax3-expressing cells was determined by counting these cells and all DAPI-labelled nuclei in the neural tube and comparison made between wild type (four embryos, six sections each) and *Slc7a5*-null (three embryos, six sections each), each dot represents the average for one embryo, unpaired *t*-test, *P* = 0.018 (see original source data). Error bars indicate SEM.

Source data are available online for this figure.

low-level transcription of these ISR-linked genes in these cell populations in wild-type embryos, further suggest that this initially adaptive mechanism normally operates in dynamic cell populations to ensure cell homeostasis.

## *Slc7a5* loss triggers the integrated stress response via phosphorylation of GCN2 and eIF2α and induces apoptosis

The canonical ISR pathway is activated via phosphorylation, and so, inactivation of the translation initiation factor eIF2α by one or more of four eIF2α kinases and amino acid deprivation triggers activation of the kinase GCN2 [23]. Increased phosphorylation of GCN2 and eIF2α was detected in E9.5 *Slc7a5*-null whole embryo lysates (Fig 6A and B). Moreover, we were able to detect localised induction of p-eIF2α in E9.5 *Slc7a5*-null embryos, extensively in the forebrain and locally in key tissues including the otic vesicle (Fig 6C–f2). These findings indicate that loss of *Slc7a5* triggers activation of the kinase GCN2 and further substantiate the induction of ISR in *Slc7a5*-null embryos.

Induction of ATF4 targets *Trib3, Chac1* and *CHOP* is indicative of a progressed cell stress response [46,47], which if unresolved ultimately leads to apoptosis [52,58]. Here, we confirmed raised levels of Trib3 protein in *Slc7a5*-null embryos (Fig 6G), which correlate with the localised high-level transcription of this gene (Fig 5F–H). We therefore next assessed cell death in *Slc7a5*-null embryos using a TUNEL assay. This revealed widespread apoptotic figures in the developing brain of E9.5 *Slc7a5*-null embryos (Fig 6H–L); however, this varied between samples and the profound and varied morphological defects in this region in null embryos made it difficult to quantify with respect to wild type (Fig 6I). Analysis of TUNEL-positive cells in sections of the dorsal spinal cord and caudal hindbrain also revealed variability between embryos (with an F-test *P* = 0.0003), but could be quantified (Fig 6J and K, quantified in L). These findings indicate that most *Slc7a5*-null embryos exhibit increased cell death. This is consistent with local triggering of apoptosis as stress levels rise on a cell-by-cell basis.

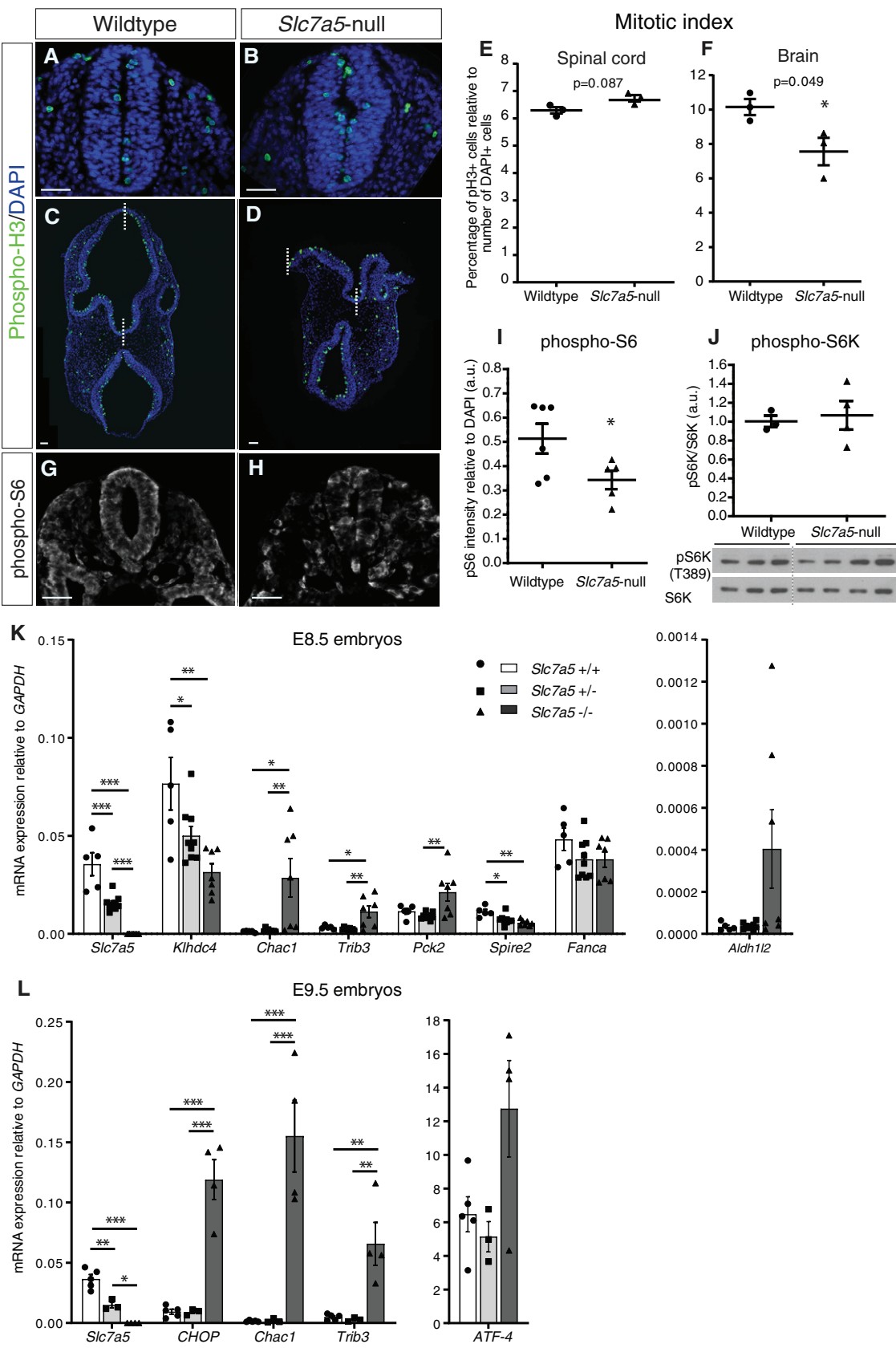

**Figure 4.**

Figure 4. **Slc7a5-null neural tube exhibits slightly reduced cell proliferation and aberrant mTORC1 activity, and qPCR validation of significantly changed genes in Slc7a5-null embryos.**

A–D  Proliferation was assessed in the spinal cord (forelimb level) (A, B) using a phospho-Histone 3 (phospho-H3) antibody (green) to identify mitotic cells and DAPI (blue) to label nuclei; cells were counted in 3 wild-type (18 sections) and 3 *Slc7a5*-null embryos (18 sections) and mitotic index calculated; (C, D) proliferation was also assessed in the forebrain in three wild-type embryos (17 sections) and three *Slc7a5*-null embryos (16 sections). White dashed lines indicate midline.

E, F  Comparison of mitotic index in wild-type and *Slc7a5*-null (E) spinal cord, *P* = 0.872, and (F) forebrain *P* = 0.049, each dot represents the average for one embryo, unpaired *t*-test (see original source data).

G, H  mTORC1 activity in the spinal cord (hindlimb level) was assessed by immunofluorescence using an antibody against the phospho-ribosomal protein S6 (phospho-S6 Ser 235/236).

I  Labelling intensity was measured and plotted relative to DAPI intensity; six wild-type embryos (18 sections) and five *Slc7a5*-null embryos (19 sections). Each dot represents the average for one embryo, (*P* = 0.0457) unpaired *t*-test with Welch correction (see original source data).

J  Western blot of individual E9.5 wild-type (*n* = 3) nd *Slc7a5*-null (*n* = 4) embryo lysates immunoblotted using an antibody against phospho-S6K (Thr389) and total P70S6K for loading control. Band intensities were measured with FIJI, and an unpaired *t*-test was performed for statistical analysis (see original source data).

K, L  (K) Validation by qPCR of the targets identified by RNA-seq; qPCR was performed on individual E8.5 embryos of each genotype (wild type *n* = 5, heterozygous *n* = 9 and *Slc7a5*-null *n* = 7). (L) qPCR was performed on individual E9.5 embryos (wild type *n* = 5, heterozygous *n* = 3 and *Slc7a5*-null *n* = 4) using primers specific for genes associated with the integrated stress response. A one-way ANOVA test with a Tukey post-test was performed.

Data information: *P < 0.05, **P < 0.01, ***P < 0.001, for actual *P*-values, see original source data file. Error bars indicate SEM. All scale bars 50 μm.
Source data are available online for this figure.

**Table 1. Seven genes significantly changed in E8.5 Slc7a5-null embryos.**

| Gene_ID | Gene | logFC | P-value | FDR | Description |
|---|---|---|---|---|---|
| ENSMUSG00000110631 | Gm42047 | −11.1 | 0 | 0 | lincRNA |
| ENSMUSG00000040010 | Slc7a5 | −4.95 | $<10^{-16}$ | $<10^{-16}$ | Solute carrier family 7 (cationic amino acid transporter, y+ system), member 5 |
| ENSMUSG00000074052 | BC048644 | −4.35 | $4.6 \times 10^{-12}$ | $4.0 \times 10^{-8}$ | cDNA sequence BC048644 |
| ENSMUSG00000112478 | AC153370.2 | −3.59 | $<10^{-16}$ | $2.8 \times 10^{-14}$ | lincRNA |
| ENSMUSG00000040263 | Klhdc4 | −1.14 | $<10^{-16}$ | $<10^{-16}$ | Kelch domain containing 4 |
| ENSMUSG00000010154 | Spire2 | −0.948 | $3.1 \times 10^{-11}$ | $2.4 \times 10^{-7}$ | Spire homolog 2 (Drosophila) |
| ENSMUSG00000032815 | Fanca | −0.387 | $7.2 \times 10^{-6}$ | 0.038 | Fanconi anaemia, complementation group A |
| ENSMUSG00000040618 | Pck2 | 0.594 | $7.3 \times 10^{-10}$ | $4.8 \times 10^{-6}$ | Phosphoenolpyruvate carboxykinase 2 (mitochondrial) |
| ENSMUSG00000032715 | Trib3 | 1.851 | $6.9 \times 10^{-8}$ | 0.00040 | Tribbles pseudokinase 3 |
| ENSMUSG00000027313 | Chac1 | 3.431 | $1.2 \times 10^{-13}$ | $1.2 \times 10^{-9}$ | ChaC, cation transport regulator 1 |

RNA-seq data based on 5 wild-type littermates and 5 Slc7a5-null E8.5 embryos identified 7 genes showing significant change in expression (and see Appendix Table S1). *Slc7a5* was downregulated with a log2 fold change of 4.948 (equivalent to > 99.9% reduction in mRNA expression (Fig 4H)). *Klhdc4* (Kelch domain containing 4) and *Spire2* (spire-type actin nucleation factor 2) genes were both downregulated to a much lesser extent with log2 fold changes of ~ 1, confirmed by reductions in mRNA expression to ~ 50% of wild-type level. *Klhdc4* is a highly abundant transcript, and *Spire2* is functionally redundant with *Spire1* (Pfender *et al*, 2011; Fig 4H), so the observed changes in their expression levels seem unlikely to be functionally significant (and may be associated with their close proximity to the *Slc7a5* locus on chromosome 8). Upregulated genes, *Chac1*, *Trib3* and *Pck2*, are all metabolic stress-related genes (see text). Correlation between samples was high (*r* > 0.98), and additional analysis without one WT outlier had a small effect on the significant gene list, adding several further genes associated with metabolic stress (Appendix Table S2).

### Slc7a5 expression depends on Wnt/β-catenin signalling, and loss of this pathway induces the stress response gene Trib3

Finally, we noticed that the *Slc7a5*-null phenotype is similar to that observed in Wnt pathway mutants, which also exhibit defects in the dorsal neural tube, involving neurogenesis and neural crest as well as in limb development [1,2,59]. Moreover, there is a good correspondence between Wnt signalling and *Slc7a5* expression in caudal epiblast/primitive streak and dorsal neural tube as well as in the limb bud [60] (Fig 1b3, b6, c3, E). Dorsal neural tube defects are dependent on canonical, β-catenin-mediated, Wnt signalling [61]. This suggests that loss of such signalling might contribute to the *Slc7a5*-null phenotype and/or that *Slc7a5* transcription is normally promoted downstream of β-catenin signalling.

To address whether canonical Wnt signalling is compromised in *Slc7a5*-null embryos, we assessed expression of the β-catenin transcriptional target *Axin2*. This revealed robust expression of *Axin2* in both littermate (Fig 7A, a1–a3) and *Slc7a5*-null embryos (Fig 7B, b1–b3) indicating that loss of *Slc7a5* does not impact canonical Wnt signalling. We next investigated whether *Slc7a5* is a potential canonical Wnt signalling target, first by analysing its promoter region for binding sites of relevant transcription factors using the eukaryotic promoter database and the MatInspector program [62]. These analyses identified two putative LEF1 and TCF7L2 sites upstream of the mouse *Slc7a5* transcription start site (TSS) and multiple potential MYC binding sites flanking the TSS (Fig EV4 and see Dataset EV3), supporting the possibility that *Slc7a5* is a target of this pathway in the developing embryo.

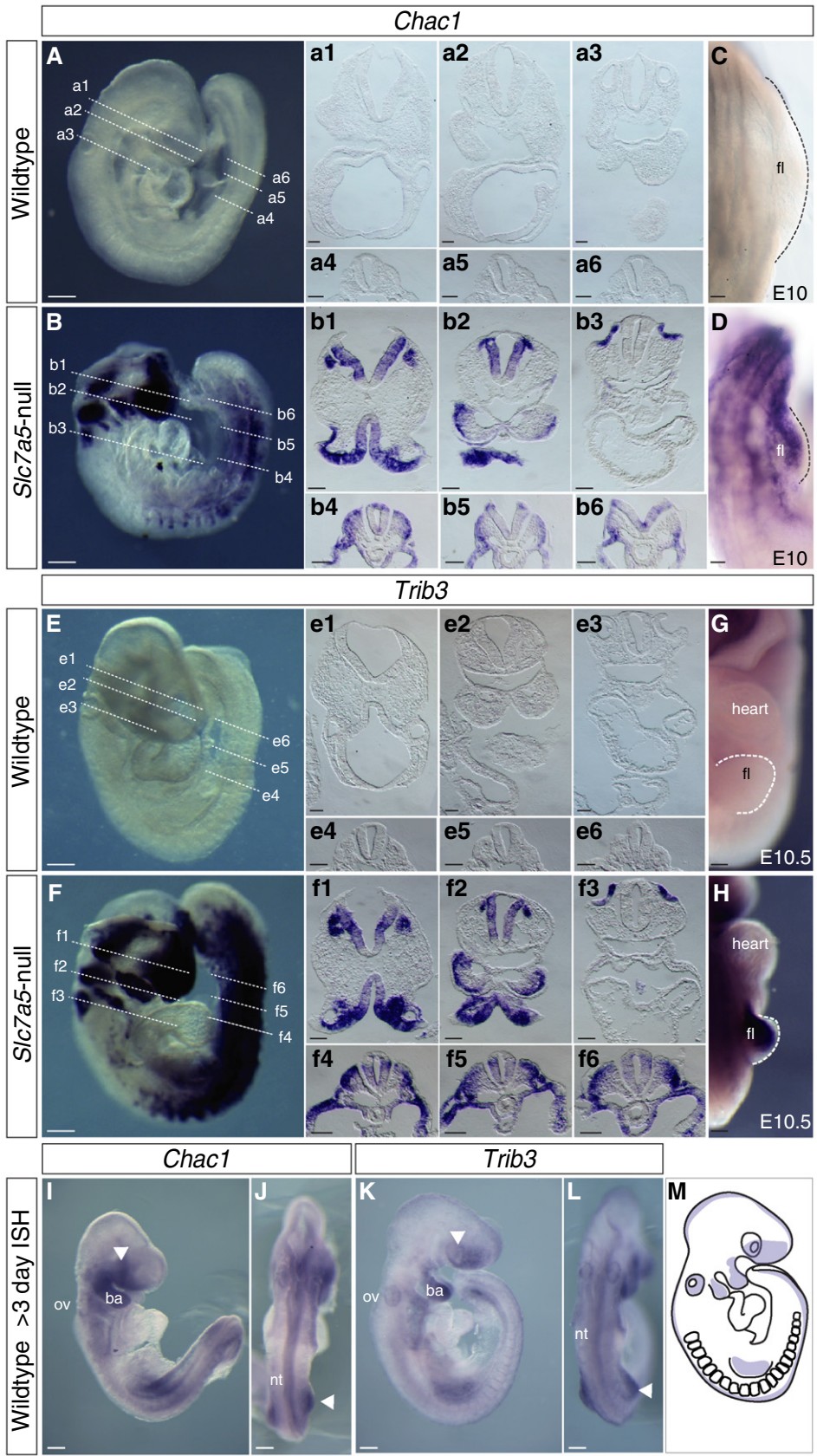

**Figure 5.**

**Figure 5.** *Slc7a5*-null embryos exhibit localised integrated stress response.

A–L   mRNA *in situ* hybridisation was performed at E9.5 to detect *Chac1* in (A–a6) wild-type and (B–b6) *Slc7a5*-null littermate embryos (n = 3 for each condition); *Chac1* expression at E10 in emerging limb buds assessed in (C) wild-type and (D) *Slc7a5*-null littermate embryos (n = 2 for each condition); *Trib3* mRNA detected in (E-e6) wild-type and (F–f6) *Slc7a5*-null littermate embryos (n = 3 for each condition) and in emerging limb buds at E10.5 assessed in (G) wild-type and (H) *Slc7a5*-null littermate embryos (n = 2 for each condition). *Chac1* and *Trib3* transcripts were also detected in wild-type E9.5 embryos after > 3 days *in situ* hybridisation reaction (n = 3 and n = 6, respectively). Lateral view (I and K), optic vesicles (white arrowheads), branchial arches (ba) and otic vesicles (ov). Dorsal view (J and L), neural tube (nt) and limb buds (white arrowheads). Scale bars 200 μm for whole embryo images and 100 μm for sections.

M   Schematic indicating regions (purple) in which high levels of *Slc7a5* (see Fig 1C–E) and ISR genes *Trib3* and *Chac1* are detected in wild-type embryos.

To test this regulatory relationship, embryo trunk explants were exposed to the Wnt secretion inhibitor Wnt-c59, which inhibits Wnt palmitoylation [63]. After 16 h, *Axin2* expression was unchanged in DMSO vehicle-only condition, but repressed by Wnt-c59 (Fig 7C, c1, D, d1). Similarly, *Slc7a5* was attenuated in the presence of Wnt-c59 (Fig 7E, e1, F, f1), placing *Slc7a5* transcription downstream of Wnt signalling. We then assessed *Slc7a5* expression in mouse embryos lacking functional β-catenin. As deletion of β-catenin results in early embryonic lethality [64], we used a conditional approach, crossing mice harbouring floxed alleles of β-catenin (Ctnnb1[tm2Kem]) [65] with a mouse line expressing Cre recombinase under the control of the T (Brachyury) promoter (T-Cre) [66] and mutant embryos generated are referred to here as *T-Cre;Ctnnb1[flLOF/Δ]*. In these mice, the Cre recombinase is expressed in axial progenitors and their descendants which give rise to the posterior spinal cord and paraxial mesoderm [66–68]. This cross-generates embryos in which β-catenin is lost from the developing body axis beginning in the caudal epiblast and primitive streak from ~ E7.5 and by E9.5 also in axial progenitor descendants. At E9.5, *T-Cre;Ctnnb1[flLOF/Δ]* embryos exhibit a truncated phenotype in which development of paraxial mesoderm is attenuated and remaining axial tissue forms neuroepithelium [68]. Importantly, this caudal-most neural tube lacked *Slc7a5* transcripts compared to littermate controls (Fig 7G, g1–g3 and H, h1–h3), indicating that *Scl7a5* transcription in the neural tube relies on β-catenin.

As loss of β-catenin leads to failure to form paraxial mesoderm, it is formally possible that the absence of other signals provided by this tissue underlies loss of *Slc7a5* expression in the adjacent neural tube. If this were the case, we would expect *Slc7a5* to be lost in embryos in which the β-catenin downstream transcription factors *Sp5* and *Sp8* are mutated, as these embryos also fail to form paraxial mesoderm [69]. However, as in littermate controls (Fig EV5A, a1–a3), *Sp5/Sp8* compound mutant embryos continue to express *Slc7a5* right to the caudal end (Fig EV5B, b1–b3). This finding indicates that *Slc7a5* expression in the neural tube does not depend on signals from the newly formed paraxial mesoderm and further that *Slc7a5* is regulated by β-catenin downstream transcription factors other than Sp5 and Sp8. Indeed, available ChIP-seq data show that while Sp5 and Sp8 bind to early mesodermal genes, they do not target *Slc7a5* in differentiating mouse ESCs [70].

As *Slc7a5* loss induces the ISR, we next assessed whether β-catenin deletion also triggers this stress response. Analysis of early (E7.5–8.25) *T-Cre;Ctnnb1[flLOF/Δ]* embryos revealed mosaic induction of ISR gene *Trib3* in the earliest T-Cre domains, including posterior epiblast and primitive streak, but not in littermate controls (Fig 7I–j2). Later at E9.5, *T-Cre;Ctnnb1[flLOF/Δ]* embryos, but not littermate controls, exhibited patches of *Trib3*-expressing cells in the region of the body axis truncation (Fig 7K–m3). This included groups of cells

where somitic mesoderm should have formed (see open arrows Fig 7m2) and in gut endoderm (a known site of T-Cre expression [66]) (Fig 7l3) as well as cells in the lumen of the spinal cord (Fig 7l2); this pattern of later *Trib3* induction may reflect mosaic Cre recombination and time required for the consequences of *Slc7a5* loss to be manifest.

Together, these findings show that Wnt and β-catenin signalling are required for *Slc7a5* transcription in the forming neural tube and implicate canonical Wnt signalling in constraint of the integrated stress response.

## Discussion

By elucidating the expression pattern, requirement and regulation of the amino acid transporter *Slc7a5*, this study uncovers a mechanism by which cell metabolism is regulated by developmental signalling in the mammalian embryo. Elevated expression of *Slc7a5* in tissues undergoing morphogenesis revealed that amino acid transport is patterned in the embryo. The phenotype of *Slc7a5*-null embryos indicated a requirement for increased amino acid transport for neural tube closure, neurogenesis and neural crest development as well as limb bud outgrowth. Aberrant mTORC1 activity and neural progenitor proliferation may underlie aspects of this phenotype. However, pre-phenotypic transcriptomics and subsequent detection of ATF4 transcriptional targets as well as phosphorylated GCN2 and eIF2α indicated *Slc7a5* loss rapidly triggers the ISR. Moreover, this correlated with elevated apoptosis, which is the likely cause of the observed developmental defects. The detection of low levels of ISR gene transcripts in wild-type embryos further suggested that cells participating in morphogenesis are particularly vulnerable to cell stress. Finally, the requirement for Wnt signalling for *Slc7a5* expression identified a new regulatory mechanism, in which Wnt promotes morphogenesis at least in part by inducing expression of this essential amino acid importer, thereby locally supporting metabolic demand and forestalling cell stress during this critical developmental period.

These data show that the effects of *Slc7a 5* loss in the mouse embryo involve rapid induction of the ISR and are most likely due to intracellular LNAA deficiency activating the eIF2α kinase, GCN2. Indeed, ISR genes including *Trib3*, *Aldh1l2* and *Pck2* are among the top transcripts induced by leucine starvation in HEK293 cells [71]; moreover, *Slc7a5* loss in human cancer cell lines has been shown to induce this stress response [19,72,73]; we detect aberrant activity of the amino acid sensor mTORC1; and transcriptomic analysis did not reveal changes in other potentially linked pathways, dopamine and thyroid hormone systems. While we detect elevated phosphorylation of GCN2 and eIF2α indicative of ISR induction in *Slc7a5*-null

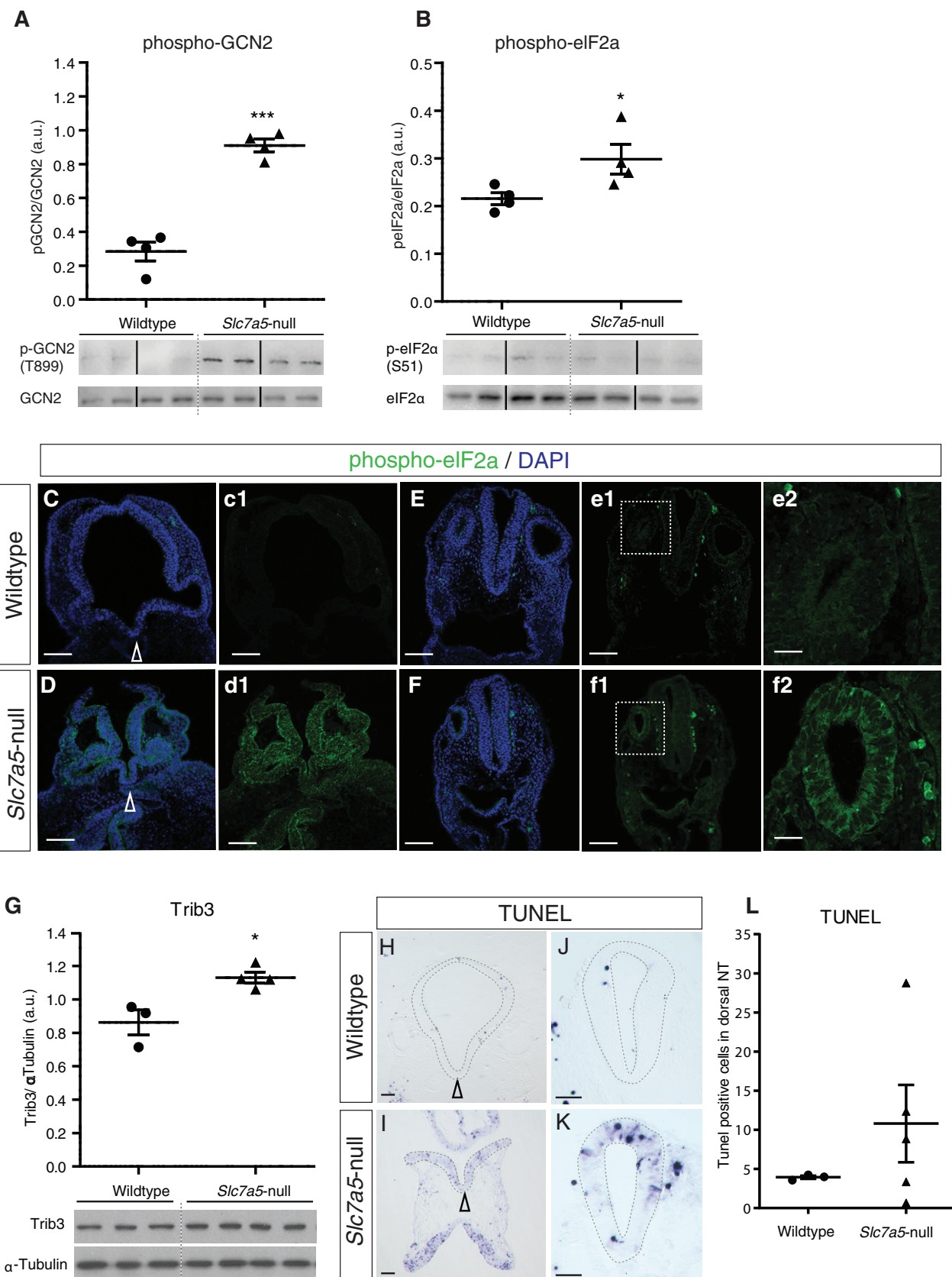

**Figure 6.**

**Figure 6.  Cell death is increased in *Slc7a5*-null embryos, and exogenous ER stress induction in wild-type embryos upregulates *Chac1* and *Trib3*.**

A, B   Western blot of individual E9.5 wild-type (*n* = 4) and *Slc7a5*-null (*n* = 4) embryo lysates immunoblotted using an antibody against phospho-GCN2 (Thr899) and total GCN2 (loading control), ***P* = 0.0001; or against phospho-eIF2α (Ser51) and total eIFα (loading control), *P* = 0.0482, band intensities measured with FIJI and analysed with unpaired *t*-test (see original source data).

C–F   Phosphorylated eIFα was assessed by immunofluorescence in E9.5 embryos in the forebrain, wild type (C, c1) (*n* = 2 embryos, 0/14 sections) and *Slc7a5*-null (D, d1) (*n* = 2 embryos 16/16 sections), and in hindbrain/anterior spinal cord, wild type (E–e2) (*n* = 4 embryos, 0/62 sections) and *Slc7a5*-null (F–f2) (*n* = 4 embryos, 54/54 sections) including in the otic vesicle (compare e2, f2). Scale bars in (C–e1, D–f1) 100 μm and in (e2 and f2) 25 μm. Arrowheads indicate ventral midline.

G     An increase in Trib3 protein was detected by Western blot in *Slc7a5*-null embryos; each lane represents an E9.5 embryo lysate (wild type *n* = 3, *Slc7a5*-null, *n* = 4) with α-tubulin loading control, unpaired *t*-test *P* = 0.0153 (see original source data).

H–L   TUNEL assay to detect apoptotic cells in wild type (*n* = 3 embryos, 32 sections) or *Slc7a5*-null (*n* = 5 embryos, 47 sections); (H, I) transverse sections through the head at level of the forebrain and (J, K) the spinal cord, (H–K) scale bars 50 μm, arrowheads indicate ventral midline. (L) Quantification of TUNEL-positive cells within spinal cord sections was performed as stated in Materials and Methods. Each dot represents the average apoptosis cell count for a single embryo, Welch's correction unpaired *t*-test, *P* = 0.250, *F*-test to compare variances, *P* = 0.0003 (see original source data).

Data information: All error bars indicate SEM.
Source data are available online for this figure.

embryos, progression of the ISR was indicated by induction of the pro-apoptosis gene *CHOP,* which together with ATF4 directly promotes *Trib3* [47,74] and *Chac1* [46]. The correlation between *Trib3* and *Chac1* induction and increased apoptosis demonstrated that these genes are good reporters for ISR activity in the embryo and allowed identification of tissues particularly sensitive to cellular stresses. This included developing neural tissue, which we additionally confirmed in response to exogenous ER stressors in E8.5 embryos, but also limb buds which form later in E9.5 embryos and exhibit high-level *Chac1* and *Trib3* in *Slc7a5*-null embryos. Together, these data strongly suggest that rapid induction of apoptosis underlies the developmental defects generated by *Slc7a5* loss and identify ISR induction as a potential upstream driver of this process. Interestingly, transcriptional targets for further cell death pathways identified in the context of ER stress, such as that mediated by IRE1 [75], were not detected in our E8.5 RNA-seq data, but studies at later stages are required to exclude such activity. To confirm ISR causality, rescue experiments are also required: although we note that inhibiting this stress response will not compensate for the underlying amino acid deficiency and so may attenuate but will not resolve this phenotype.

The appearance of a phenotype in *Slc7a5*-null embryos only at E9.5 indicates that this gene is not critically required for some

morphogenesis events that take place during early embryogenesis. This includes gastrulation and axial elongation and may reflect functional redundancy with the related gene, *Slc7a8/Lat2* (although this is not fully reciprocal; the *Slc7a8* knockout mouse has only a minor phenotype [76]). Nonetheless, loss of amino acid import by both these transporters (a condition generated when their regulatory subunit, *CD98/Slc3a2*, is mutated [77–79]) results in earlier lethality from E7.5 [14]. Death of *Slc7a5*-null embryos at E10.5 may reflect onset of dependence on chorion–allantoic circulation and *Slc7a5* expression in extra-embryonic tissue and later placenta [80,81].

Importantly, the detection of low-level *Trib3* and *Chac1* transcripts in *Slc7a5* domains of expression in wild-type embryos suggested that adaptive ISR is normally ongoing in energetic cell populations in the mammalian embryo. This is consistent with a homeostatic role for elf2α phosphorylation in buffering protein synthesis in normal unstressed cells [82]. *Slc7a5* is one of a number of stress-mitigating genes involved in amino acid transport reported to be upregulated by ER stress in cell lines [83,84]. Indeed, the *Slc7a5* gene contains a cis-acting ER stress response element (ERSE) that binds ATFs [83,85]. These findings suggest that *Slc7a5* expression is normally regulated by positive feedback once the ISR is activated. Such action may contribute to elevated *Slc7a5* expression during energetic activities and serve to ensure timely morphogenesis.

**Figure 7.  Wnt β-catenin signalling is unaffected in *Slc7a5*-null embryos, but is required for *Slc7a5* transcription and β-catenin loss induces ISR gene *Trib3*.**

A, B   (A, a1) Wnt β-catenin target *Axin2* strongly expressed in littermate, (a2, a3) in TS, and (B, b1) in *Scl7a5*-null embryos (b2, b3) in TS (*n* = 2 littermates, *n* = 6/6 *Slc7a5*-null embryos).

C, D   (C, d1) *Axin2* transcripts in embryo trunk explant exposed to DMSO (C, c1, *n* = 12/12) were reduced in Wnt-c59-exposed explants (D, d1, *n* = 14/14). Images in (c1, d1) show TS through explants (data from three independent embryo explant experiments).

E, F   *Slc7a5* transcripts in embryo trunk explant exposed to DMSO (E, e1, *n* = 9/10) or Wnt-c59 (F, f1, *n* = 15/15). Images in (e1, f1) show TS through explants (data from 3 independent embryo explant experiments).

G, H   *Slc7a5* expression in control *T-Cre;Ctnnb1^flLOF/Δ^* heterozygous littermate embryo (G, g1) and in TS (g2, g3), and in *T-Cre;Ctnnb1^flLOF/Δ^* mutant embryo where *Slc7a5* transcripts are absent in posterior neural tube (the region of T-Cre recombination, indicated with white arrowhead in H) and shown in TS in (h2, h3) (*n* = 0/6 littermate and *n* = 6/6 mutant embryos).

I, J   *Trib3* is lacking in *T-Cre;Ctnnb1^flLOF/Δ^* heterozygous control embryos at E7.75 (I), caudal region dissected from the same embryo (white dashed box in I) (i1) and viewed in TS (i2); *Trib3* detected in homozygous *T-Cre;Ctnnb1^flLOF/Δ^* embryo (J), in caudal region dissected from the same embryo (white dashed box in J) (j1) and shown in TS in caudal epiblast and remnant primitive streak (ps) (j2) (*n* = 3/3 *T-Cre;Ctnnb1^flLOF/Δ^* embryos and 14 littermate controls). *Trib3* was also detected in the more rostral forming neural tube at this early stage (seen in J) and so in cells not directly experiencing β-catenin loss; this may reflect the failure to form paraxial mesoderm, which provides signals that promote and support neurogenesis.

K–M   At E9.5, *Trib3* was not detected in *T-Cre;Ctnnb1^flLOF/Δ^* heterozygous littermate embryos (K, k1) and in TS (k2, k3), but was detected in homozygous *T-Cre;Ctnnb1^flLOF/Δ^* mutant embryos (L-m3) in patches of cells in neural tube (nt) (L-l2), gut (g) (l3) and somites (s) (m2, arrowheads) (*n* = 9/9 *T-Cre;Ctnnb1^flLOF/Δ^* embryos and 12 littermate controls).

Data information: White dashed lines indicate position of sections. Scale bars, in (A, a1, B, b1, G, g1, H, h1) = 200 μm, in (C–F) = 100 μm, all sections 50 μm.

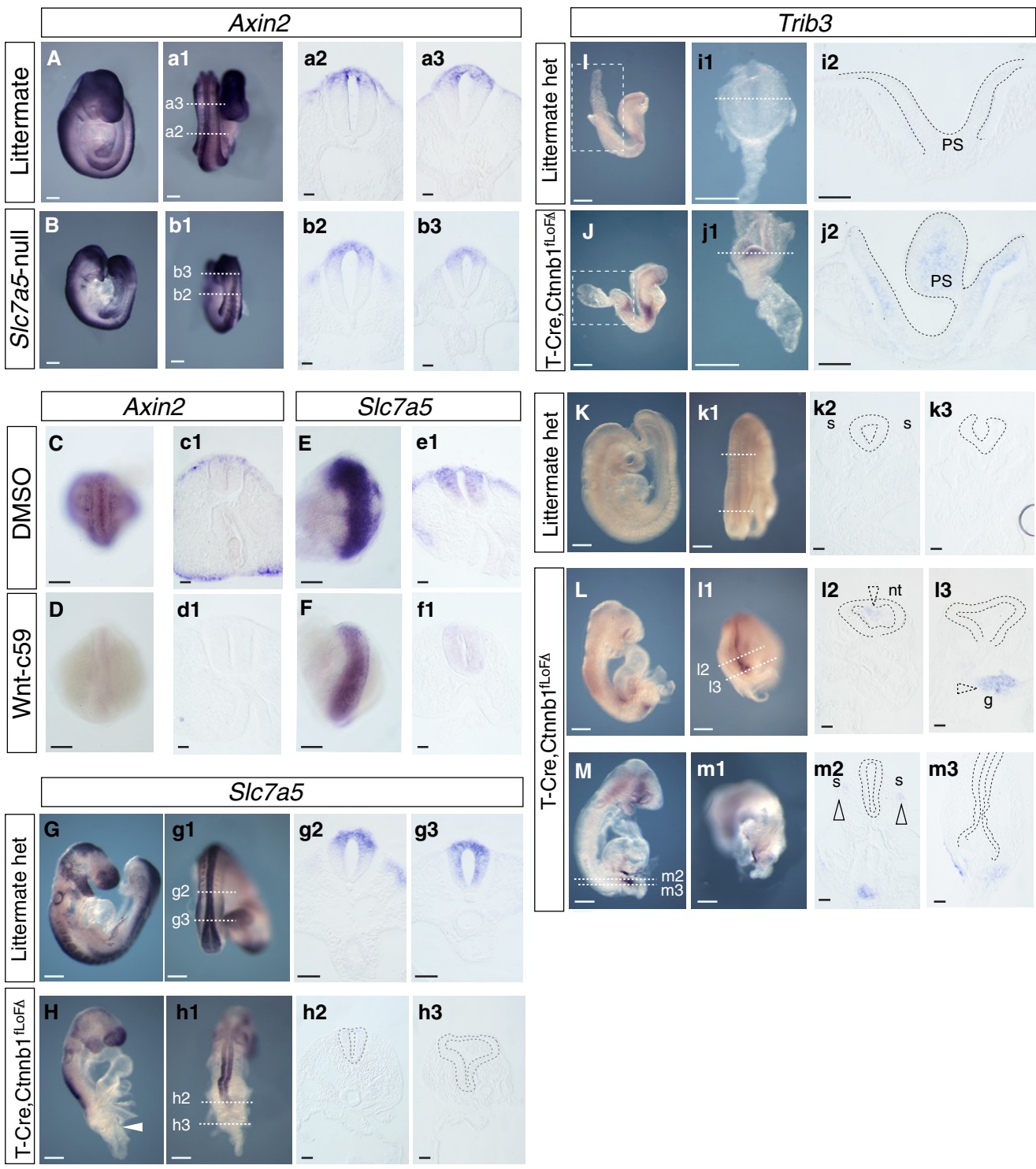

**Figure 7.**

Although *Slc7a5* may be upregulated by adaptive ISR in the embryo, this seemed unlikely to be its only regulator as mice lacking this stress pathway do not phenocopy the *Slc7a5* deletion [82]. The similarities between the phenotypes of *Slc7a5*-null and Wnt pathway mutants are striking. Not only do they affect many of the same cell populations, with Wnt mutants also exhibiting defects in

neurulation, neural crest [59] and neurogenesis [61] as well as limb bud outgrowth [1,2], but they also share phenotypic detail. For example, in the dorsal neural tube, loss of *Slc7a5* or Wnt reduces neuronal differentiation [61] and both *Wnt3a/Wnt1* double-mutant [59] and *Slc7a5*-null embryos possess cell populations derived from the first emerging neural crest, but exhibit reduction of later forming

dorsal root ganglia, consistent with a failure to expand the Pax3 expressing neural crest progenitor pool [42]. Importantly, such dorsal neural tube defects have been shown to result specifically from loss of β-catenin-mediated signalling [61] and also Myc [42] and we show here that *Slc7a5* transcription is lost following inhibition of Wnt secretion or deletion of β-catenin in the forming neural tube. Furthermore, the presence of binding sites for known mediators of canonical Wnt signalling LEF/TCF and the direct Wnt target MYC upstream of *Slc7a5*, along with ChIP-seq data, showing that β-catenin binds the *Slc7a5* promoter in the *Xenopus* embryo [86] and ChIP-PCR data demonstrating that MYC directly binds the *Slc7a5* promoter in human cancer cell lines and promotes *Slc7a5* transcription [19] strongly support the possibility that *Slc7a5* is a direct Wnt-β-catenin/MYC target in the mouse embryo. This conclusion is reinforced by screens which place Wnt upstream of *Slc7a5* in other cellular contexts and point to conservation of this regulatory relationship [87,88].

In addition, Wnts have other downstream pathways and these include promotion of mTORC1 activity in a variety of cell lines, independently of β-catenin signalling [89–91]. As loss of *Slc7a5* and so local amino acid deficiency lead to aberrant mTORC1 activity, this suggests a further explanation for the similarity between mutations affecting Wnt ligands [59] and *Slc7a5*-null embryos. Indeed, mTORC1 activity is required for normal neurogenesis in the chick neural tube [92] and for neural crest differentiation and survival [93]. This also suggests operation of cross-talk between mTORC1 and β-catenin-mediated Wnt pathways, which serves to maintain proteostasis. Such regulation may in turn be modulated by input from the adaptive ISR as it promotes expression of stress-mitigating genes including *Slc7a5*. Moreover, Wnt promotion of metabolic gene expression is characteristic of many cancer cell states, for example, β-catenin-promoted cMyc expression upregulates glycolytic genes in such contexts [94], supporting the view that multiple Wnt downstream pathways orchestrate cellular metabolism. Our finding that the ISR gene *Trib3* is induced in β-catenin mutant embryos further connects Wnt signalling to the regulation of cellular stress.

Importantly, monitoring in *Slc7a5*-null embryos for phospho-S6 and also TUNEL-positive cells underlined that the consequences of stress in tissues are manifest on a cell-by-cell basis and that these may differ according to the activity of individual cells during embryogenesis. Moreover, a critical mass of cells may need to be affected to generate a developmental defect and while adaptive stress may delay morphogenesis, chronic stress and so transition to apoptosis may be more likely to occur when multiple stresses are present. Indeed, the ISR integrates a range of cellular stresses which activate distinct eIF2α kinases; for example, amino acid deficiency activates GCN2 and ER stress kinase PERK (protein kinase R (PKR)-like endoplasmic reticulum kinase), while viral or bacterial infection activates protein kinase R (PKR) and iron deficiency activates haem-regulated kinase (HRI) by [23]. It may be that small changes in, for example, nutrient supply could promote sustained cellular stress across a cell population in particular genetic backgrounds or when pathogens abound. Indeed, there are examples of gene–environment interactions which account for the sporadic nature of human diseases, such as Crohn's [95], neural tube [18,96] and heart, skeletal and renal defects [97], as well as those induced by hypoxia [98,99]. Moreover, hypoxia-induced reduction in FGFR1 protein leading to heart defects in mouse embryo has been linked

to induction of p-PERK and the ISR in the context of the unfolded protein response [100].

These findings make important new links between signalling, metabolism, cell stress and developmental defects in the mammalian embryo and suggest that Wnt activity normally locally promotes *Slc7a5* to elevate amino acid supply in cells participating in morphogenesis. This identification of Wnt signalling as a key regulator of cell stress in embryonic tissues opens the way to wider investigation of how metabolic demand and the ISR are regulated in particular cell populations and how the level, timing and integration of cell stresses lead to specific developmental defects.

# Materials and Methods

### Mouse lines and embryo collection

*Slc7a5*$^{fl/fl}$ mice harbouring two copies of the *Slc7a5*-targeted allele (exon 1 of *Slc7a5* flanked with two loxP sites) were crossed with a mouse line ubiquitously expressing Cre recombinase under the Bal1 promoter (*Bal1*-cre) to obtain a global *Slc7a5* knockout mouse line [16]. Heterozygous *Slc7a5*$^{+/-}$ C57Bl/6 mice were viable and fertile and were bred free of Bal1-cre in subsequent generations [16]; however, *Slc7a5*$^{+/-}$ inter-crosses did not produce any *Slc7a5*-null live mice. Here, heterozygous *Slc7a5*$^{+/-}$ mice were crossed to generate litters between E8.5 and E11.5. An abnormal phenotype was not found in E8.5-null embryos (35/128 *Slc7a5*$^{-/-}$ embryos from 18 litters), but was apparent at E9.5 (35/162 *Slc7a5*$^{-/-}$ embryos from 19 litters) and the oldest live *Slc7a5*-null embryos were found at E10.5 (6/28 *Slc7a5*$^{-/-}$ embryos from three litters). Sex and genotype of embryos were determined by PCR.

To generate embryos lacking β-catenin in axial progenitors, Ctnnb1$^{tm2Kem}$ mice [65] harbouring two copies of the floxed β-catenin loss-of-function (LOF) allele (exons 2–6, including the ATG and domains essential for binding E-cadherin and TCF/LEF flanked with two loxP sites), referred to as *Ctnnb1*$^{flLOF/flLOF}$, were crossed with a mouse line homozygous for the *T-Cre* transgene (which expresses Cre recombinase under the control of the *T/Bra* promoter [66]) and heterozygous at the *Ctnnb1* locus deleted for exons 2–6 (*T-Cre*$^{tg/tg}$; Ctnnb1$^{\Delta ex2-6/+}$). In this cross, both mutant (*T-Cre*$^{tg/+}$; Ctnnb$^{\Delta ex2-6/flLOF}$) (referred to here as *T-Cre;Ctnnb1*$^{flLOF/\Delta}$) and control littermates (*T-Cre*$^{tg/+}$; Ctnnb1$^{flLOF/+}$) (referred to here as littermate het) are generated at 50% frequency. *Sp5/Sp8* compound mutants were generated as described in [69].

Wild-type CD1 embryos were collected between E7.0 and E9.5. Embryos were fixed in PFA 4% and processed for *in situ* hybridisation or immunofluorescence as described below. Mouse colonies were breed, mice sacrificed and embryos isolated following Home Office guidelines (PPL 60/04454 or PPL 60/3455 and/or 60/4118).

### Mouse embryo and explant culture and exposure to small molecules

Live E8.5 CD1 embryos were dissected from the uterus and collected within yolk sacs in warm (37°C) culture media (F12 + Glutamax + FCS 10%). Hanging drop culture method [101] was then used with DMSO (1:1,000) or ER stress inducers: tunicamycin (1 μg/ml) and thapsigargin (1 μM). Embryos were cultured for 6 h

at 37°C in 5% $CO_2$, then dissected, fixed and processed for *in situ* hybridisation. E9.5 CD1 mouse embryos were dissected in ice-cold medium, and explants of the trunk region were made, embedded in collagen and cultured in OptiMem/10%FBS/1×GlutaMAX/1× B27/1xPen strep (all from Gibco) in an incubator with 5% $CO_2$ at 37°C, as described previously [102]. Explants were cultured in either vehicle control DMSO or Wnt-C59 (Tocris, Cat. No. 5148) at 4 μM (1/2,500) in DMSO for 16 h. Explants consisted of full thickness of the embryo at the level of the three most recently formed somites (caudal explant) or at the level of the three next rostral somites (rostral explant) (caudal explants were only compared with other caudal explants and rostral explants only compared with other rostral explants). Following fixation in ice-cold 4% PFA, explants were processed for *in situ* hybridisation to detect mRNAs or for immunocytochemistry as described above.

### *In situ* hybridisation for mRNA

*In situ* hybridisation experiments were performed on whole embryos to detect mRNA for *Delta1, Neurog2, FoxG1, Fgf8, Hes5, Sox10, Slc7a5, Slc3a2, Chac1* and *Trib3* following standard procedures. Primers used to clone *Slc7a5, Slc3a2, Chac1* and *Trib3* can be found in Appendix Table S1. Subset of embryos subjected to whole-mount *in situ* hybridisation was embedded and cryo-sectioned using standard procedures to localise mRNA at a cellular level.

### Immunofluorescence

A standard protocol was used to embed and cryo-section embryos at 20 μm for immunofluorescence. Primary antibodies were used at indicated concentrations: phospho-H3 (S10), phospho-S6 (S235/236) and phospho-S6 (S240/244) (1:500—Cell Signaling Technology #9706, #5364, #2211), phospho-eIf2α (1:500—Cell Signaling Technology #3398), tubulin-βIII (Tuj1, 1:500—MMS-435P, BioLegend), Pax3 (1:200—supernatant DSHB) and FoxA2 (1:200—ab108422, Abcam). Secondary antibodies goat anti-rabbit IgG (H+L) cross-adsorbed secondary antibody, Alexa Fluor 594 (A-11012, Thermo Fisher) and donkey anti-mouse IgG (H+L) secondary antibody, Alexa Fluor 488 conjugate (A-21202, Thermo Fisher) were incubated for 30 min at 1:500 at room temperature. Nuclei were stained with DAPI.

### TUNEL assay

Detection of cells with DNA-strand breaks was performed on sections of E9.5 wild-type or *Slc7a5*-null embryos by the TUNEL labelling method, using an ApopTag kit (Millipore) according to the manufacturer's instructions. Images were taken with Leica DMRB microscope using a ×20 objective. To identify TUNEL-positive cells, a colour threshold of 100 was set in ImageJ, and all cells above this threshold were counted in the dorsal 700 μm of the spinal cord. For quantification, see text, figure legends and original source data.

### Microscopy

Whole embryo images (mRNA *in situ* hybridisation and Tuj-1 immunofluorescence) were taken using a Micropublisher 3.3RT and Q-Imaging on a Leica MZFLIII dissecting microscope. Representative

sections were imaged using Leica DRB compound microscope. On sectioned tissues, immunofluorescence was imaged using either a personal DeltaVision imaging system (20× lens for Figs 3C–F, 4A, B, G, H and Appendix Fig S2A–D; 10× lens Fig 4C, D (nine images taken and stitched using ImageJ) or with SP8 Leica confocal microscope ×10 (Fig 6C–f1) ×60 (Fig 6e2, f2). To quantify the varied pattern of phospho-S6 (Fig 4G, H, I), fluorescence intensity was measured using ImageJ and normalised relative to the intensity of DAPI fluorescence in the same area (see original source data). Figures were assembled using Adobe Photoshop and Illustrator software for composite figure construction.

### Preparation of RNA-seq samples

E8.5 embryos with 8–10 somites were dissected in cold PBS and lysed individually with TRIzol reagent. Embryos were stage matched with respect to somite number, and only male embryos were used to avoid potential sex-related variations in transcriptome. Allantois was kept for genotyping and sexing the embryos by PCR [103]. Total RNA was extracted using the RNeasy mini kit (Qiagen). Samples were DNase-treated "on-column" using RNase-free DNase (Qiagen). Quality and quantity of total RNA were measured using a Qubit fluorometer (Invitrogen). Following quality control, 5 wild-type and 5 *Slc7a5*-null male embryos were selected as biological replicates. Samples started with 300 ng total RNA per embryo to which spike-ins (ERCC, 2 μl of 1:100 dilution) were added. We used a TruSeq RNA v2 kit (Illumina RS-122-2001) to prepare the cDNA library following the manufacturer's instructions; polyA RNA selection was performed and fragment and random priming followed by cDNA synthesis, ligation and ×10 rounds of PCR amplification. RNA-seq was performed by Illumina HiSeq 2000 with 100 bp paired ends and "dye-swap" spike-in of all samples. This gave paired-end reads, length 101 bp, between 28 and 42 million reads per sample.

### Computational analysis of RNA-seq data

Reads were aligned to the Ensembl GRCm38 primary assembly genome (release 90) using *STAR* ver. 2.5.3a [104] with the following parameters –*genomeLoad NoSharedMemory –outSAMstrandField intronMotif –outSAMmode Full –outFilterMultimapNmax 2 –outFilterMismatchNmax 5 –outFilterType BySJout –outSJfilterIntronMaxVsReadN 5000 10000 15000 20000 –readFilesCommand zcat –outSAMtype BAM SortedByCoordinate –outSAMunmapped Within –outReadsUnmapped Fastx –quantMode GeneCounts*. Read counts per gene were calculated in the same *STAR* run. The gene annotation file (release 90) was downloaded from Ensembl. For differential expression, edgeR ver. 3.18.1 [105,106] was applied with its default normalisation, which resulted in a list of $\log_2$ fold changes, raw *P*-values and Benjamini–Hochberg-corrected *P*-values to control the false discovery rate (FDR) (see Table 1, Appendix Tables S1 and S2).

### Quantitative PCR

RNA from E8.5 or E9.5 embryos (respectively, 200 and 500 ng) was reverse-transcribed using the qScript cDNA synthesis kit (Quanta Biosciences). A 1/10 dilution of the cDNA was used to perform qPCR on a Bio-Rad iCycler with PerfeCTa SYBR Green

          

FastMix (Quanta Biosciences). Primer pairs used can be found in Appendix Table S2. qPCRs for each embryo were run in triplicate, and the average of these technical replicates was taken to represent one independent experiment. Comparisons of such data from at least three embryos (see Figure legends) were made for each condition and each gene of interest. All data were determined using ΔΔCt method [107] and shown relative to GAPDH with error bars indicating ±SEM. A one-way ANOVA with a Tukey post-test was performed for statistical analyses (see original source data).

**Preparation of whole embryo lysates and Western blotting**

To prepare samples for SDS–PAGE electrophoresis, isolated whole mouse embryos were immediately transferred to a clean microtube containing ice-cold lysis buffer [50 mM Tris–HCl, pH 7.5, 1% Triton X-100, 1 mM EGTA, 1 mM EDTA, 150 mM NaCl, 0.27 M sucrose, 50 mM sodium fluoride, 10 mM sodium 2-glycerophosphate, 5 mM sodium pyrophosphate, 1 mM sodium orthovanadate, 1 mM benzamidine, 1 mM PMSF and 0.1% 2-mercaptoethanol, EDTA-free protease inhibitor cocktail (Roche) (1 individual embryo per tube)]. Following sonication and vortexing, the resulting homogenate was centrifuged at 3,000 *g* for 10 min at 4°C to remove remaining tissue debris, and the resulting supernatant was used for Western blot analysis. Each lane was loaded with lysate from one embryo (this was 20 μg, except for the blot for Slc3a2 in which lanes were loaded with 25 μg). Loaded lysates were then subjected to SDS–polyacrylamide gel electrophoresis and immunoblotted as previously described [16] using primary antibodies (1:1,000) against (Cell Signaling, Beverly, MA, USA, phospho-P70S6K (T389) #9205, P70S6K #2708, phospho-eIF2α (S51) #3398 and eIF2α #9722, TRB3 (Proteintech, Chicago, IL, USA, #13300-1-AP) and α-Tubulin (Sigma, St Louis, MO, USA, #T6074, used at 1:5000), pGCN2 (T899) (Abcam ab75386) and GCN2 (Abcam ab134053), Slc3a2 (Santa Cruz #sc-20018), b-Actin (Cell Signaling Technology cat#4970). Primary antibody detection was carried out with (Cell Signaling, Beverly, MA, USA) secondary antibodies, anti-rabbit IgG-HRP #7074 or anti-mouse IgG-HRP-linked #7076 at (1:5,000) as appropriate by ECL. Resulting band intensities were quantified using FIJI software (NIH, Bethesda, MD), and unpaired t-tests were performed for statistical analyses. In figures, solid black bars indicate where blots were cut and dashed lines separate wild-type and *Slc7a5*-null lanes. See original source data for complete blots.

## Data availability

RNA-seq data are available in ArrayExpress accession number E-MTAB-6336, https://www.ebi.ac.uk/arrayexpress/experiments/E-MTAB-6336/.

**Expanded View** for this article is available online.

## Acknowledgements

We thank members of the Storey Laboratory and colleagues Cheryll Tickle, Andy Copp, Linda Sinclair and David Ron for critical reading of versions of this manuscript. We are grateful to Dr Ravi Chalamalasetty for generation and collection of *T-Cre;Ctnnb1^{flLOF/Δ}* and *Sp5/Sp8* compound mutant embryos. This research was supported by a Wellcome Trust project grant to PT (WT094226)

on which KGS was a named collaborator and NP was further supported for a short period by the ISSF (WT204816/Z/16). YBS is supported by NIH, and TPY is supported by NIH (project number 1ZIABC010345-18). This study utilised microscopy resources supported by a Wellcome Trust multi-user equipment grant for the development of tissue imaging approaches (WT101468). KGS is a Wellcome Trust Investigator (WT102817AIA).

## Author contributions

Conception or design of the work or the acquisition, analysis or interpretation of data for the work: NP, MG, MF, CL, PAH, GAS, AD, TPY, PMT and KGS; Drafting the work or revising it critically for important intellectual content: NP, MG, MF, CL, PAH, TPY, PMT and KGS; Final approval of the version to be published: NP, MG, MF, CL, PAH, GAS, AD, Y-BS, PMT and KGS; Agreement to be accountable for all aspects of the work in ensuring that questions related to the accuracy or integrity of any part of the work are appropriately investigated and resolved: KGS.

## Conflict of interest

The authors declare that they have no conflict of interest.

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
