## [Review Process File · EMBO Reports]

Wnt regulates amino-acid transporter *Slc7a5* and so constrains the integrated stress response in mouse embryos

Nadège Poncet, Pamela A. Halley, Christopher Lipina, Marek Gierliński, Alwyn Dady, Gail A. Singer, Melanie Febrer, Yun-Bo Shi, Terry P. Yamaguchi, Peter M. Taylor and Kate G. Storey

Review timeline:

Submission date:	13 May 2019
Editorial Decision:	14 May 2019
Revision received:	30 August 2019
Editorial Decision:	8 October 2019
Revision received:	18 October 2019
Accepted:	25 October 2019

Transaction Report: This manuscript was transferred to *EMBO reports* following peer review at *The EMBO Journal*

1st Editorial Decision

14 May 2019

Thank you for the transfer of your research manuscript and the associated referee reports from The EMBO Journal to EMBO reports. As discussed, we would like to invite you to revise your manuscript for potential publication in EMBO reports.

I note that all three referees, who evaluated your study for The EMBO Journal, considered the findings potentially interesting and I want to propose the following revision plan for EMBO reports:

- It will not be necessary to provide causality, i.e., evidence that the induction of the ISR causes the phenotype of *Slc7a5*-deficient mice, for publication in EMBO reports. It will be sufficient to carefully phrase your conclusions and to acknowledge that the observations are correlative, even though such an explanation might be likely.

- Please address the following points in the revision:

Please provide evidence that the *Slc7a5* probe is specific and provide data on *Slc3A2* protein expression levels (referee 1). Please extend the data on proliferation by providing an assessment of the mitotic index (referee 1 and 2). The analysis of mTORC1 activity and ISR activation should be completed by providing stainings using an antibody against the S240/244 phosphosite of S6 and data on the phosphorylation of GCN2 and PERK (referee 1).

The data on stress-sensitive cell populations (all referees, Figure 6) should be moved to the Supplement. If possible, luciferase experiments could address the direct role of Wnt signalling components (Myc, Mad, Max) in the regulation of *Slc7a5*. Otherwise, please tone down and carefully phrase your conclusions and acknowledge that a direct role has not been shown.

I look forward to seeing a revised version of your manuscript when it is ready. Please let me know if you have questions or comments regarding the revision.

Authors' point by point response to Referee comments (in blue):**Referee #1:**

The manuscript by Poncet et al focuses on the role of the large neutral amino acid transporter *Slc7a5* during mouse embryogenesis. The authors investigate the expression of *Slc7a5* in wild type mice and interrogate the phenotype of *Slc7a5* knockout embryos. They show that morphogenesis is severely disrupted in the *Slc7a5* knockout embryos with resulting effects on mTORC1 activity as well as an integrated stress response transcriptional signature. Finally, they demonstrate that the Wnt signaling pathway is required for *Slc7a5* expression.

The ideas presented in the manuscript are interesting, however there are multiple issues with how the data are interpreted and the conclusions formulated. Moreover, evidence for the conclusions drawn in this manuscript need to be strengthened with additional experiments.

Major :

1) Figure 1 :

Authors nicely showed the mRNA expression pattern of *Slc7a5* during embryogenesis. However, authors should perform the same *in situ* hybridization experiment using the *Slc7a5* knockout embryos as control for staining specificity. *We have carried out ISH for *Slc7a5* transcripts *Slc7a5* null embryos alongside their littermates as positive controls. We did not detect a signal in *Slc7a5* null embryos. This data is now presented in new Appendix Figure S1.*

In addition, a recent study demonstrated that disruption of *SLC7A5* expression leads to a strong decrease in *SLC3A2* expression in cancer cells (Cormerais et al. 2016). As *Slc3a2* is significantly downregulated in the RNAseq data from table S1, authors need to control if *Slc3a2* expression is maintained in *Slc7a5* *-/-* embryos.

Table S1 shows that *Slc3A2* is close but not significantly downregulated (FDR=1, it is ranked at 52). We also note that *SLC3A2* additionally binds *SLC7A8*, *SLC7A7*, *SLC7A6* and *SLC7A11*, (Bröer S. and Fairweather SJ. *Compr. Physiol.* 2018 Dec 13;9(1):343-373) but none of these genes are significantly changed in the RNAseq data (*Slc7a11*, rank 64 logFC 1.336; *Slc7a7*, rank 6393 logFC 0.328; *Slc7a8*, rank 14042 logFC 0.2; *Slc7a6*, rank 19203 logFC -0.19). Cormerais et al found that *SLC3A2* (also known as CD98) downregulation did not inhibit *Slc7a5* activity in cancer cells, with low CD98 levels being sufficient for normal growth and mTORC1 activity in cancer cells. These findings suggest that reduction in *SLC3A2*/CD98 is unlikely to be responsible for the phenotype of *Slc7a5* mutant embryos.

However, to investigate this possibility we first carried out Western blots on *Slc7a5* mutant and wildtype embryos. In our initial experiments using two different antibodies (sc-390154 Mouse anti CD98 (F-2) and Biorad VPA00372 Rabbit anti CD98) we were unable to detect this protein in either mutant or wildtype embryos. We then tried a third anti-CD98 antibody (sc-20018-s Rat anti CD98, H202-141), this had a high background but provided a faint band at the predicted position, with no significant difference between wildtype and mutant embryos (Appendix Figure S3).

Given that the comment here was prompted by interpretation of *Slc3a2* transcript levels in the RNAseq data set (generated with pre-phenotypic E8.5 embryos) we also cloned *Slc3a2* (from NM_001161413.1, probe region is 920-1922 - 3'end of coding sequence) and carried out *in situ* hybridisation for this gene in *Slc7a5*-null (n=3), heterozygous (n=2) and wildtype (n=2) embryos. This revealed strong expression of *Slc3a2* in all embryos, which were assessed at E9.5 when the phenotype is apparent (Appendix Figure S3).

These new Western blot data together with *Slc3a2* *in situ* data and the finding of Cormerais et al that even low levels of CD98 protein are sufficient for *Slc7a5* activity and tumour growth, indicate that reduction of *Slc3a2*/CD98 is unlikely to be responsible for the *Slc7a5*-null phenotype.

This is of primary importance for the conclusions made in this study as *Slc3a2* has been reported to not only interact with AA transporters, but also to regulate integrin signalling (Feral et al. PNAS 2007), which is also involved in embryogenesis.

We note that *SLC3A2* aka *CD98hc*^{-/-} embryos die shortly after implantation (Tsumura H et al. 2003 Biochem Biophys Res Commun. 2003 Sep 5;308(4):847-51). *CD98hc* mutants (Δ *CD98hc*- β geo) that can mediate integrin signalling but cannot support amino acid transport through *SLC7A5*, *SLC7A8*, *SLC7A7* and *SLC7A6* die between E7.5 and E9.5 (Sato et al. 2011 Cell Biosci. 2011 Feb 25;1(1):7). Our *Slc7a5*-null embryos die only after E10.5 suggesting that they are not subject to an immediate loss of *CD98* and integrin signalling loss, which would lead to earlier death and this conclusion is supported by the new data provided in Appendix Figure S3.

2) Figure 4 :

Line 178 - Authors used an antibody against phosphor-H3 in order to compare the mitotic index in the spinal cord region between WT and *Slc7a5* KO. Although no significant change has been detected, authors argue in the text that this experiment revealed a slight reduction of the mitotic index in the absence of *SLC7A5*. As this experiment resulted in no significance, this is an overstatement and has to be modified. The term slight reduction was meant to imply that this is only a tendency, the p value was 0.056. Would this be grounds to dismiss any effect (see referee 2 pt1)? Our complete statement says: "This revealed a slight reduction in mitotic index in the absence of *Slc7a5* (Fig. 4C). This modest effect suggests that reduced cell proliferation is unlikely to be the major explanation for the defects observed in *Slc7a5*-null neural tube." Moreover, we chose to assess proliferation in a region of the mutant neural tube that had completed neurulation so that we could compare most easily with control embryos – however, this is a region with the least dramatic phenotype.

We have now clarified this issue by analysing the mitotic index in the forebrain which exhibits more profound phenotypic defects and also re-assessing this in the spinal cord (our data for latter was generated again so that the same imaging and cell counting regime could be applied to data from both regions) (revised Figures 4A-F). These new data show that there is no difference in mitotic index in mutant embryos in the spinal cord and an only just significant difference in the forebrain. We think that this supports the view that reduced proliferation is unlikely make a major contribution to the defects observed, which may reflect earlier induction of the ISR and apoptosis (see text and new Figure 6).

fig 4D-G - Authors compared mTORC1 activity between WT and *SLC7A5* KO mice using western blot and immunochemistry. Additional mTORC1 targets should be investigated by western blotting such as 4EBP1 and S6 in order to have a complete investigation of the mTORC1 pathway. Moreover, results between western blot and immunochemistry show a discrepancy. While no difference is detected in S6K phosphorylation using western blot, immunochemistry shows a decreased S6 phosphorylation in the preneural tube. Authors explain this difference arguing that changes in mTORC1 signalling only happens in a subset of cells and is therefore not detectable in a whole embryo lysate. However, the authors used an antibody against the S235/236 phosphorylation sites of S6 which are not mTORC1 specific. Previous studies demonstrated that the MAP kinase pathway can phosphorylate the S235/236 independently of mTORC1 activity (Roux et al. JBC 2007). Therefore, the authors should repeat this experiment using an antibody against the S240/244 phosphorylation site of S6 and confirm that the MAPK pathway is not altered between WT and *Slc7a5* embryos.

We have established in our embryo explant assays that the phospho-S6 S235/236 detected with this antibody is greatly reduced on exposure to rapamycin (mTORC1 inhibitor) and we can provide this data. The issue here is whether its reduction in the *Slc7a5*-null embryos similarly reflects attenuation of mTORC1 activity detected in these embryos or if it reflects loss of MAPK signalling (assume this means ERK1/2 activity). pERK1/2 is not detected in the neural tube at the stages when these experiments are carried out (Corson et al Development. 2003 Oct;130(19):4527-37) so this seems unlikely. However, to directly address this, we have used an antibody against the S240/244 phosphorylation site of S6 and compared levels in wildtype and *Slc7a5* mutant embryos. This revealed a similar variable pattern of phospho-S6 across tissues in *Slc7a5*-null embryos compared with wildtype controls (Appendix Figure S2).

Figure 4H-I and Figure 6F-G:

RNAseq and qPCR data from figure 4H-I suggests an integrated stress response signature in the *Slc7a5* KO embryos. To confirm these data, the authors investigated the activity of the ISR by

measuring p-eIF2/eIF2 ratio and Trib3 expression by western blot. Indeed, the phosphorylation of eIF2 is highly dynamic and is sometimes hard to catch especially if this change happens in small cell population. However, the present data are not enough to clearly demonstrate the activation of the ISR in the *Slc7a5*^{-/-} embryos. The author should complete these figures by measuring the phosphorylation level of GCN2 and PERK as well as the expression level of ATF4 as they are other indicators of ISR activation. Moreover, the same immunofluorescence approach used with phospho S6 in figure 4 should be performed with phospho eIF2. Finally, in addition to Trib3, both CHOP and Chac1 were found upregulated in the RNAseq data. Therefore, western blot for these two proteins is required to confirm the conclusions made by the author.

In the paper, we already present qPCR data that validate RNAseq data for ISR genes *ATF4*, *CHOP*, *Chac1* and *Trib3* (the latter 3 are all known to be ATF4 transcriptional targets) which we show are upregulated in the *Slc7a5* mutants (revised Figure 4L). Moreover, we demonstrate local ectopic high level induction of both *Chac1* and *Trib3* by mRNA *in situ* hybridisation in the embryo, showing too that this takes place in regions/tissue that normally express *Slc7a5*.

As requested by the EMBO reports editor we have now run Western blots for phospho-GCN2 and phospho-PERK. We show in new figure 6A that phospho-GCN2 levels are significantly increased in *Slc7a5*-null embryos. Phospho-PERK was not consistently detected in mutant embryos. We have also repeated the Western blot analysis for phospho-eIF2 α with a further set of wildtype and *Slc7a5*-null embryos and this is now just significant (new Figure 6B). In addition, to our surprise, we were able to detect regionally localised phospho-eIF2 α in *Slc7a5*-null embryos by immunofluorescence (revised Figures 6C to f2).

3) Figure 6H-K: The relevance of this experiment in the context of the study is not clear. As *Slc7a5* is an amino acid transporter and its deletion is more likely to result in amino acid starvation rather than ER stress. The author should perform the same type of experiment using either a media depleted of LNAA or using the *Slc7a5* inhibitor JPH203. We do understand that loss of *Slc7a5* (an amino acid transporter) is more likely to result in amino acid starvation rather than ER stress. The aim of the experiment with ER stressors was to determine whether the cell populations particularly vulnerable to cell stress in the embryo at this time were the same regardless of the initial trigger. We compared the cell populations experiencing ER stress (using ISR genes *Trib3* and *Chac1* as readouts) with those affected in *Slc7a5* mutants. This revealed a similar pattern of ISR activation (particularly affecting the neural tube) indicating that at specific times in development certain tissues are particularly sensitive to stress (triggered by either ER stress or AA deficiency). The proposed experiment with JPH203 is an interesting suggestion, although time required to deplete AA and so trigger the ISR may be longer than the culture period possible in this hanging drop embryo assay (~5h). In the context of this paper, we think that this experiment will not add significantly to the conclusions drawn already from analysis of *Slc7a5* mutant embryos. As requested by the EMBO reports editor we have moved these experiments with ER stressors to a new Appendix Figure S6.

4) Figure 7G-H :

As *Slc3a2* is required for *Slc7a5* activity, authors should investigate if *Slc3a2* expression is also dependent on the Wnt/B-catenin pathway. This too is an interesting suggestion, but we feel it is beyond the scope of this study.

Figure 7 and S9 :

In the figure 7G-H, the authors demonstrate that *Slc7a5* transcription in the neural tube relies on β -catenin signalling. In figure S9, the authors demonstrate that β -catenin deletion induces *Trib3* transcription in the region of the body axis truncation including groups of cells where somatic mesoderm should have formed, in gut endoderm and cells in the lumen of the spinal cord. According to these results, the region where β -catenin deletion leads to *Trib3* expression does not correlate with the region where *Slc7a5* is downregulated by β -catenin deletion (neural tube). These results suggest that induction of the stress protein *Trib3* due to β -catenin deletion is independent of the *Slc7a5* downregulation. The conclusions stated here and the title saying that the Wnt pathway regulates *Slc7a5* to constrain the ISR in mouse embryo are both misleading as these events happen in different tissues.

There is some misunderstanding here about the cell populations affected by T-Cre mediated loss of β -catenin, the expression pattern of *Slc7a5* and the timing of induction of *Trib3* expression. We appreciate that this requires familiarity with cell populations and lineages in the developing embryo and we apologise that this was not explained in greater detail in the paper.

T-Cre is first active in caudal lateral epiblast cells (where *Slc7a5* is expressed, Figures 1B, b5, b6) and this is where we first detect some *Trib3* expressing cells in T-Cre β -catenin mutant embryos (E7.5-E8.5) (Figs 7I-j2). Cells in this epiblast are axial progenitors which will give rise to the neural tube and paraxial mesoderm and some cells continue on to form rudiments of these tissues in T-Cre/ β -catenin mutant embryos. Cre recombination is well known to be variable in onset and in timing of consequence between cells and we therefore assessed for *Slc7a5* and *Trib3* expression in embryos at a later stage E9.5 (Figs 7G-h3 and Figs 7K-m3 respectively). At E9.5 some neural and some paraxial tissue has formed before axial truncation then takes place. This most recently formed neural tube lacks *Slc7a5* expression (Figs 7G-h3; also see loss of *Slc7a5* in embryo explants treated with Wnt secretion inhibitor, figs 7C-f1) and *Trib3* is detected as the tissue undergoes apoptosis and cells accumulate in the lumen (Figs L, l2). This indicates that time needs to elapse between loss of *Slc7a5* transcription (and consequent protein and finally AA depletion) in these cells before *Trib3* is induced. Similarly, in the paraxial mesoderm we see induction of *Trib3* in cells which would have formed somites, again this is triggered later in this cell population (Figs 7M, m2). So, we do conclude from these experiments that Wnt/ β -catenin signalling is required for *Slc7a5* expression; and that a downstream consequence of loss of β -catenin is also the induction of *Trib3*, and this is likely to reflect loss of *Slc7a5* (as *Trib3* is dramatically and locally upregulated in *Slc7a5* mutant embryos, Figs 5A-H), but it takes longer to achieve AA depletion downstream of β -catenin loss. As we note in the discussion, while Wnt promotion of *Slc7a5* contributes to metabolic support for energetic cells in the developing embryo and is required to constrain the ISR (as loss of *Slc7a5* triggers the ISR), Wnt/Myc activity regulates other metabolism genes in other contexts and *Slc7a5* may be one of several Wnt regulated metabolism genes operating in developing embryos. Consistent with this (and as noted in the paper) T-Cre is also expressed in the gut endoderm (Perantoni et al 2005, Development 132: 3859-71) and loss of β -catenin under T-Cre leads to *Trib3* induction in this tissue too.

Minor :

- line 155 and 156, authors refer to figs 3O,P,R while it appears to be figs 2O,P,R, please modify. This has been corrected

- Studies demonstrating induction of the ISR following SLC7A5 disruption has been published previously and should be cited (Rosilio et al. Leukemia 2015 ; Cormerais et al. Cancer Res 2016). These findings in cancer cell lines are now cited in the first paragraph of the Discussion.

Referee #2:

In this manuscript, Poncet and colleagues characterize the expression pattern and the loss of function phenotype of *Slc7a5*, an Na⁺-independent amino acid transporter involved in the delivery of large neutral amino acids to cells. They describe the expression pattern of *Slc7a5* during mouse embryogenesis in detail, and show that *Slc7a5* null mice have developmental defects in a number of tissues. They further show that the Integrated Stress Response mediated by ATF4 is activated in the mutant embryos, and provide evidence that Wnt and beta-catenin genes are required for proper *Slc7a5* expression.

This manuscript could be subdivided mainly into three parts: (1) Expression pattern analysis and the loss of function phenotype of *Slc7a5*. This part is descriptive in nature, but otherwise, the data is solid. (2) Characterization of mTORC1 and ISR signalling in *Slc7a5* null embryos. While most of the data here are solid, the authors make over-interpretations on a number of points. For example, the authors argue that "ISR is likely the underlying cause of the phenotype," without any supporting data to back up this claim (see below for details). (3) A possible role of Wnt signalling in *Slc7a5* induction during embryogenesis. The authors do provide some data to support this idea, but there are also some conflicting data. This section requires further validations and clarification. Below are a few specific major concerns along these lines:

1. The authors show in Figure 4C that phosphor-H3 positive cells are slightly reduced, and they write in page 9 that "reduced proliferation is unlikely to be the major explanation for the defects observed in *Slc7a5*-null neural tube." What is the basis of this conclusion? Unless there is quantitative reasoning, I don't think one can simply dismiss reduced cell proliferation as a

contributing phenotype. This comment is the opposite to that of referee#1. The reduced cell proliferation we observed did not reach significance but was borderline. As we note above (see referee 1, point 2) we have now further addressed this by assessing mitotic index in the forebrain, a region of the embryo with more profound phenotype and compared this with the mitotic index in the spinal cord as well (new Figures 4A-F). These new data show that there is not a significant difference between wildtype and *Slc7a5*-null embryos in the spinal cord and an only just significant difference in the forebrain.

2. The authors perform gene expression profiling in *Slc7a5* null embryos and find that target genes of the unfolded protein response (UPR), including *Chac1* and *Trib3*, are induced. In page 11, the authors conclude that "loss of *Slc7a5* function as a transporter of LNAA and associated induction of the ISR as the likely underlying cause of these early neural developmental and limb defects." This is an overstatement, as authors have only shown ISR induction, but did not establish ISR's causal role in developmental defects. It is well known that when the ISR is not resolved it leads to apoptosis. CHOP can promote cell death via multiple mechanisms (reviewed in Pakos-Zebrucka et al EMBO Rep. 2016 Oct;17(10):1374-1395) and *Trib3* (ATF4 target) is a major driver of apoptosis (Ohoka N et al 2005 EMBO J **24**: 1243–1255; Du K et al 2003 Science **300**: 1574–1577): we detect *CHOP* and both *Trib3* transcript and *Trib3* protein elevation in the *Slc7a5* mutants, furthermore *Trib3* is induced locally in regions where *Slc7a5* is normally expressed and in tissues that exhibited the phenotypic changes (for example, cranial ganglia, otic vesicle and limb buds, Figure 5). Indeed, we found apoptosis extensively in the brain and were able to quantify this increase in a region where the neural tube managed to close and had a comparable morphology between *Slc7a5*-null and wildtype littermates. Given the induction of ISR genes and extensive apoptosis it is reasonable to suggest ("likely") that the underlying cause of these defects is ISR induction.

Why could ISR signalling not be a mere consequence of reduced amino acid import in those tissues? In order to establish the causality of ISR signalling, the authors need to block ISR signalling and demonstrate that the embryonic phenotypes are suppressed.

This is not a trivial experiment. We had considered it. The small molecule ISRIB would have to cross to the embryo via the placenta and this may not be possible in time to rescue the phenotype that appears at E9.5 (as the placenta is only just beginning to be established at this time). An alternative would be to expose embryos in hanging drop culture to ISRIB and see if this reduces apoptosis. The window of operation here is ~ 5h so effects may be difficult to establish in this timeframe. Following discussion with EMBOJ and EMBO reports editors we agreed to simply clarify this issue in the text (see Discussion, p18 end of first paragraph).

3. Tunicamycin and Thapsigargin treatment experiments in Figure 6 do not add any new information, as *Chac1* and *Trib3* are already known targets of the ISR/UPR. Therefore, Figure 6H - k could either be removed, or moved to the Supplementary data. The aim of this experiment was not intended to prove that *Chac1* and *Trib3* are targets of the ISR but to localise cell populations expressing ISR genes. It confirmed that this was the case in the mouse embryo and most importantly, it revealed a similar pattern for *Trib3* and *Chac1* -indicating that at this time in development specific cell populations are sensitive to cell stress (please see response to referee 1, point3 above). This data has now been moved to Appendix Figure S6.

4. In Figure 7, the authors show evidence that beta-catenin is required for *Slc7a5* expression in the caudal most neural tube. While this result is consistent with the authors model that Wnt signalling induces *Slc7a5*, it is also possible that the phenotype is due to beta-catenin's other roles, such as in cell-cell adhesion. Experiments which demonstrate that *Slc7a5* expression is lost when Wnt ligand secretion is inhibited (with small molecule Wnt-c59) (Figures 7C-f1) appear to have been overlooked. These experiments demonstrate that Wnt-c59 attenuates known canonical Wnt signalling target *Axin2* as well as *Slc7a5* in our assay. They demonstrate that loss of Wnt at the top of the pathway (leaving β -catenin intact) also leads to loss of *Slc7a5*. The β -catenin loss of function experiments dissect the Wnt pathway further and indicate that *Slc7a5* is regulated downstream of the canonical Wnt signalling pathway.

The authors need to provide more corroborating evidence to support their main model. Since the authors report putative Lef1 Tcf binding sites in Figure S7, how about examining *Slc7a5* expression in Lef1 Tcf mutant embryos? How about generating a reporter of *Slc7a5* expression (by fusing the regulatory element of *Slc7a5* to lacZ or GFP), and determine (1) whether it responds to Wnt

signalling, and (2) whether such response is dependent on Lef1 and Tcf7L2 binding sites? – we think the experiments with Wnt-c59 provide corroborating evidence for the claim that *Slc7a5* is a Wnt regulated gene, by demonstrating that interfering with Wnt signalling at a different level in the pathway also inhibits *Slc7a5* expression. We provide some evidence that this might be direct, in identifying relevant binding sites for TCF and MYC, but we do not claim that this means it is directly regulated (please see below), nor does this seem critical information for this study. Moreover, we do not have access to Lef1/TCF mutant embryos, but note that an *in vitro* study in human cancer cell lines (Yue et al 2017, Cell Reports, 21, 3819-3832, Figure 3) has demonstrated that *Slc7a5* is directly regulated by MYC, which is an established Wnt/ β -catenin target. The latter experiments identify a key MYC binding site in the *Slc7a5* promoter (also present in the mouse, see Fig EV4 and Appendix table S5 in revised MS), demonstrate that MYC binds the promoter and that mutation of this binding site leads to loss of luciferase reporter activity in 293T cells. These data also demonstrate that inhibition of MYC leads to attenuation of *Slc7a5* transcription in other human tumour lines (P493 and Daudi cells). We now cite and discuss this data in the Discussion.

5. If *Slc7a5* is a Wnt signalling target, mutations in beta-catenin downstream transcription factors should abolish *Slc7a* expression. But in Figure S8, the authors show that *Slc7a5* expression remains normal in Sp5 and Sp8 mutant background. Doesn't this mean that *Slc7a5* is NOT a Wnt signalling target? It is established that Sp5 and Sp8 are not the only β -catenin downstream transcription factors (Kennedy MW et al. Sp5 and Sp8 recruit β -catenin and Tcf1-Lef1 to select enhancers to activate Wnt target gene transcription Proc Natl Acad Sci U S A. 2016 Mar 29;113(13):3545-50). We apologise for not making this clearer in the paper and have revised the Results and Discussion to clarify this. The point of this experiment was to still further dissect the Wnt pathway leading to *Slc7a5* expression and also to control for loss of paraxial mesoderm in canonical Wnt pathway mutants - (paraxial mesoderm is reduced in the sp5/sp8 mutants and also β -catenin mutants, but in sp5/sp8 mutants *Slc7a5* is still expressed, so the loss of mesoderm cannot be indirectly responsible for *Slc7a5* loss in the β -catenin mutants). We think the compound sp5/sp8 mouse mutants are informative, indicating that transcription factors other than Sp5 and Sp8 mediate *Slc7a5* expression. Furthermore, we identify Myc binding sites in the promoter region of the *Slc7a5* (new Figure EV4), data which appear to have been overlooked, and inspection of supplementary data listing all predicted binding sites for this region (Table S5) also includes multiple sites for Myc co-factors MAD and MAX. This further supports the point raised in the discussion that *Slc7a5* is likely to be regulated by Myc as are other cell metabolism genes (see Discussion, last para p19, top para p20).

6. The authors write in page 16 that "available Chip-seq data show that while Sp5 and Sp8 bind to early mesodermal genes they do not target *Slc7a5*." This statement appears to me as a piece of evidence arguing against the authors' model that Wnt signaling regulates *Slc7a5*. It needs to be clarified (see response to point 5 above). Is there any Chip-seq evidence that Lef1, Tcf1 bind to *Slc7a* regulatory sequence? We cite a paper that provides evidence for this in the discussion (Nakamura Y et al (2016) Tissue- and stage-specific Wnt target gene expression is controlled subsequent to beta-catenin recruitment to cis-regulatory modules. Development 143: 1914-25). This is now also provided as Data reference.

Referee #3:

The manuscript of Poncet and colleagues addresses the role of *Slc7a5* during embryonic growth. They had previously published the embryonic lethal phenotype of *Slc7a5* deletion (Poncet 2014) and here aimed to understand its nature. *Slc7a5* mediates the cellular import of a number of amino acids including the essential amino acid leucine.

They began by demonstrating that *Slc7a5* is expressed widely, especially in rapidly expanding tissues including limb buds and the nervous system. Accordingly, loss of *Slc7a5* impaired growth of these tissues. Transcriptomic analysis by RNA sequencing revealed surprisingly few genes were altered, but the induction of *Trib3* and *Chac1* led the authors to suspect activation of the integrated stress response (ISR). The ISR is a pathway initiated when one of four different stress-sensing kinases phosphorylates eIF2 α . This inhibits translation of most mRNAs, while inducing translation of the transcription factor ATF4. The authors demonstrated localized activation of the ISR in *Slc7a5* null animals in those regions normally rich in *Slc7a5* expression. Subtly increased cell death was

also detected. Finally, wnt signalling was implicated in promoting Slc7a5 expression.

This is a well-written manuscript presenting convincing data. One strength of the paper is that it adds to a growing number of papers showing that in a variety of species the ISR can affect embryonic development. It is surprising therefore that little mention of such studies is made. A weakness of this paper is that induction of the ISR is a predictable consequence of depleting amino acid levels: leucine deficiency is a well-known and potent activator of the eIF2a kinase GCN2. Overall, this study will be of interest to those studying local nutrient usage during development. We note that this will be many cell and developmental biologists, researchers interested in cell metabolism and cell stress, as well as clinicians interested in congenital defects and their causes.

Main concerns

1. The finding of an ISR in growing tissues deprived of an adequate amino acid supply through deletion of an amino acid transporter is unsurprising. To our knowledge, this is the first report of amino acid transporter deletion induced ISR in the context of a developing mammalian embryo. The regulation of *Slc7a5* and so the ISR by Wnt signalling is a further novel finding which may inform mechanism(s) underlying stress-induced congenital defects.

2. To claim that Slc7a5 functions to "constrain the integrated stress response in mouse embryos" is rather strong. Rather than act on the ISR, Slc7a5 is more likely simply to maintain appropriate amino acid availability. A failure to do so would result in deficiency and stress. We agree, *Slc7a5* maintains "normal stress" levels by providing adequate amounts of essential amino acids, without which ISR is triggered. We propose that Wnt regulation of *Slc7a5* promotes AA transport in cells undergoing energetic activities – e.g. neural crest, closing neural tube, expanding limb bud and so forestalls the ISR. We would have liked to re-word the title to better reflect this: "Wnt regulates amino-acid transporter *Slc7a5* and so constrains the integrated stress response in mouse embryos" – however the character limit for titles has required that we shorten this to "Wnt regulates amino-acid transporter *Slc7a5* constraining integrated stress response in mouse embryos" – but have requested to have the longer more accurate version.

3. The experiments using ER stress-inducing agents are a distraction since there is little evidence of ER stress in the transcriptomic data. Instead, there is simply an ISR likely due to local amino acid starvation. The aim of the experiment with ER stressors was to determine whether cell populations particularly vulnerable to cell stress in the embryo at this time were the same regardless of the initial trigger. We compared the cell populations experiencing ER stress (using ISR genes *Trib3* and *Chac1* as readouts) with those affected in *Slc7a5* mutants. This revealed a similar pattern of ISR activation (particularly affecting the neural tube) indicating that at specific times in development certain tissues are particularly sensitive to stress (triggered by either ER stress or AA deficiency). Please see our response to reviewer 1 point 3. This data has now been placed in new Appendix Figure S6.

4. Although only a minority of the RNAseq hits (two) were from the ISR, it has been focused on exclusively. The other genes appear to have been dismissed with relatively little explanation. We were struck by the results of the RNA-Seq experiment which identified only 6 significantly changed genes (excluding Slc7a5), 3 upregulated and 3 downregulated. We do discuss each gene in the legend of Table 1 and justify there the identification of the 3 upregulated genes (*pck2*, *Trib3* and *Chac1*) which are all associated with cell stress as a reason for investigating the stress response further. Genes that were downregulated *Klhdc4*, *Spire2* and *Fanca* were assessed by qPCR (Fig. 4K). The predicted change in *Fanca* was not seen by qPCR and so this was not followed up, *Spire2* is known to be functionally redundant with *Spire 1*, while *Klhdc4* (was only reduced by half in mutants) and is a gene that has as yet no known function and so these were not followed up. The association of *Trib3*, *Chac1*, *pck2* and *Aldh1l2* (the latter just below significance) with cell stress indicated that this was a major cellular process affected by *Slc7a5* loss of function and this prompted us to explore the ISR further.

5. The link between the observed phenotype and the ISR is based on correlation. Have efforts been made to modify the ISR genetically or pharmacologically to change the phenotype of Slc7a5 deletion? Please see response to referee 2 point 2 above.

Minor point

1. Although the writing is generally excellent, the following sentence from the abstract could

probably be improved: "Similar patterns of stress response gene expression, in *Slc7a5*-null, ER-stressor exposed and (at low levels) in wildtype embryos, identified stress-vulnerability in tissues undergoing morphogenesis". This has been re-written.

2. There is little discussion of the role of the ISR during development, despite a number of recent studies being published on this topic. We apologise for this unintended omission and in particular note a recent important advance demonstrating ISR/UPR induction in the mouse developing heart in response to hypoxia (Shi et al 2016, *Development*, 143, 2561-2572). This is now cited and discussed in the Discussion (para 1 p21). We note here too recent reports of the ISR impacting chondrocyte differentiation at late stages and BMP signalling in the fly embryo.

Wang et al 2018 Inhibiting the integrated stress response pathway prevents aberrant chondrocyte differentiation thereby alleviating chondrodysplasia. *Elife*. 2018 Jul 19;7. pii: e37673.

Malzer E, Dominicus CS, Chambers JE, Dickens JA, Mookerjee S, Marciniak SJ. The integrated stress response regulates BMP signalling through effects on translation. *BMC Biol*. 2018 Apr 3;16(1):34.

Arbitrating advisor's comments:

Overall I like the links that the authors have made between loss of the amino-acid transporter *Slc7a5* during embryonic morphogenesis to aberrant expression of genes implicated in stress response and apoptosis of key tissues that form the neural tube and limb. The data that drive my enthusiasm for publication in EMBO Journal are the characterization of the phenotype and *Slc7a5* expression (Figs. 1 and 2), phospho-S6 (Fig. 4E,F), the data on induction of *Chac1* and *Trib3* (Fig. 4H,I, and 5).

The tenuous links that I see are the 1) the relationship between Wnt signaling and *Slc7a5* expression and 2) aberrant integrated stress response.

To point #1, it appears that considerable data relative to the relationship between Wnt and *Slc7a5* were included in a newer version (experiment to block Wnt secretion "with small molecule Wnt-c59", analysis of *sp5/Sp8* mutants, new supplemental data). If this new data as cited in the rebuttal are included in a revised version, then I am comfortable with making the link between Wnt and *Slc7a5* regulation. Nonetheless, this remains an oversimplification as the *Slc7a5* expression pattern in Fig. 1 differs from what would be expected of the canonical Wnt signaling genes: *Slc7a5* is expressed throughout the neural tube (not dorsally like *Wnt1*), it is not increased in the isthmus (a strong domain of Wnt signaling), not particularly higher in the tail bud, etc. The authors do not comment on this discrepancy. This is not entirely accurate, while not strongly expressed at the isthmus, *Slc7a5* is initially dorsally restricted in the forming neural tube and largely remains so as the spinal cord develops, including in neural crest (Figures 1B-C4 at E8.5, E9.5 and at E10.5 new Fig EV1). We note the lack of phenotype in the tailbud /axial elongation in the discussion and suggest this may reflect redundancy with other EAA transporters.

To point #2, there is a clear and striking increase in genes associated with stress response (ER stress and UPR; *Chac1* and *Trib3*). However, their data do not show a change in canonical ISR pathway (eIF2 α) and it seems perhaps risky to explain it away by saying the phenotype has progressed beyond the initial phase (lines 330-334).

Induction of p-GCN2 and p-eIF2 α has now been demonstrated and this is presented in the revised MS (Figure 6).

As the authors mention, ISR is induced as "an adaptive response which acts to restore cellular homeostasis by decreasing global protein synthesis whilst promoting mRNA translation for selected proteins." There is no functional data shown for a decrease in global protein synthesis or promotion of selective mRNA translation. I am not asking that the authors do such experiments - just pointing out the gap in their data and agreeing that the data are interesting but correlative. For technical reasons we have not been able to measure global protein synthesis levels in mutant embryos, but we have shown localised selective induction of known ISR target genes *Trib3* and *Chac1* (Figure 5) and also *Trib3* protein by Western blot (Figure 6G).

In general, I feel the authors have provided important new in vivo data on an interesting mouse mutant, they have explored the phenotype well, and provided mechanistic insight tying to disrupted stress response. The fact that the embryos die so early made the analysis challenging but they have brought the studies to a final conclusion. The data will be interesting to basic scientists studying

cellular metabolism, nutrient sensing, developmental biology and to those studying birth defects and clinical correlations.

With that said, the reviewers do raise important points and are more expert in the field of ER stress and integrated stress response than I am. I chime in on a few of their points as well as give some suggestions for rewording.

The authors should reword the title as they suggest to Rev 3, point 2
This has been addressed and see Referee 3 point 2.

With the data currently shown, the abstract should delete the statement that "Slc7a5-null neural tube exhibited reduced cell-proliferation" as the authors themselves equivocate on this point (see their response to Rev 1 point 2 and Rev 2 point 1). The authors propose to now assess phospho-H3 in areas with more profound defects and should do so. In the legend it is unclear what stage embryos were assessed or what position along the rostral-caudal axis was assessed. From the rebuttal: "This revealed a slight reduction in mitotic index in the absence of Slc7a5 (Fig. 4C). This modest effect suggests that reduced cell proliferation is unlikely to be the major explanation for the defects observed in Slc7a5-null neural tube."

This has been addressed see revised Figure 4 and see response to referee 1 point 2 and referee 2 point 1.

Relative to the reviewers comments, Line 98-101 would be better divided into 2 sentences or the second half used as a lead-in to the subsequent sentence. The first half states facts, whereas the second half remains more speculative. Indeed, the authors should revisit the wording of their conclusion statements in lines 98-104 of the introduction. "Here we reveal that Slc7a5 expression is elevated in cell populations undergoing morphogenesis in the mouse embryo and that this process is disrupted in Slc7a5-null embryos, identifying the integrated stress response (ISR) (Harding et al, 2003; Pakos-Zebrucka et al, 2016) as the likely cause." This paragraph at the end of the Introduction has been revised.

All 3 reviewers felt the inclusion of ER stress pharmacological data was inappropriate and for me the pattern of induced Chac1 and Trib3 expression does not reflect that seen in Slc7a5 null embryos (very different rostral-caudal areas are chosen for the tissue sections). I agree that this should be limited to supplemental data.

This is now presented in Appendix Figure S6.

2nd Editorial Decision

8 October 2019

Thank you for the submission of your revised manuscript to EMBO reports. Your manuscript was evaluated again by former referee 1 and 2; please find their reports copied below.

As you will see, the referees support publication in EMBO reports pending a careful revision of all statements regarding causality of the ISR and pending that the correlative nature of the data is appropriately acknowledged. Please avoid conclusions regarding causality of the integrated stress response in mediating the phenotype and discuss alternative explanations - such as the lack of essential amino acids. We kindly ask you to track the changes in the revised document to ensure a fast and efficient evaluation of the final revision.

From the editorial side, there are also a few things that we need before we can proceed with the official acceptance of your study.

REFeree REPORTS

Referee #1:

The additional data are convincing and the authors have satisfactorily addressed my questions.

Since the Editor noted that it would not be necessary to provide causality between the induction of the ISR and the phenotype observed in the Slc7a5-deficient mice. In my opinion the manuscript should be published in EMBO Reports.

Referee #2:

This manuscript by Poncet and colleagues reports mouse Slc7a null embryo phenotypes. Specifically, they describe (1) the Slc7a expression pattern during development, (2) report the developmental defects associated with Slc7a loss, (3) show that the Integrated Stress Response (ISR) target genes are induced in Slc7a null embryos, (4) and show that Wnt/beta catenin signaling regulates Slc7a expression. The study is mostly descriptive. The induction of ISR in cells lacking the amino acid transporter Slc7a is somewhat predictable. Whether the Slc7a loss phenotypes are due to amino acid deprivation, or alternatively due to excessive ISR signaling remains unclear. However, the authors still argue in a few sections of the manuscript that ISR signaling activity is "likely to have a causal role" in the Slc7a null phenotype. There is no data supporting such as causal relationship, and therefore, those claims need to be withdrawn. Below are a few specific examples for the authors' consideration.

1. Line 258 - 260: The authors write that "induction of the ISR is the likely underlying cause of these early neural developmental and limb defects." The authors echo such a statement in the Discussion, where the authors write that "these data strongly suggest that rapid induction of the ISR and apoptosis underlie the developmental defects by Slc7a5 loss." These are overstatements, as there is no evidence presented in this manuscript to establish causality of ISR in mediating the phenotype. On the one hand, ISR activation (GCN2 and eIF2a phosphorylation) as a mere consequence of amino acid transport defect is an outcome that is predictable based on the well-established literature. But the authors need to consider the very likely scenario that the developmental phenotypes are due to specific tissues being deprived of essential amino acids for their function, and not having anything to do with excessive ISR signaling itself. If the authors wish to implicate ISR signaling, they need to repeat the experiment in mice with impaired ISR signaling and demonstrate a rescue in phenotype.
2. Line 315: After showing that there is enhanced cell death in Slc7a5 null embryos, the authors conclude that the results are "consistent with the local induction of the ISR following Slc7a loss and the triggering of apoptosis as stress levels rise on a cell by cell basis." I suggest taking out the inference of ISR in apoptosis induction here, as there could be many other causes of cell death. The authors should be mindful that there are many other models of UPR stress-induced cell death that does not involve ATF4/CHOP, which includes the role of IRE1-mediated caspase activation (Upton et al., Science 2012 PMID 23042294), Ca/calmodulin-induced apoptosis (Timmins et al., J. Clin. Invest. 2009 PMID 19741297), and IRE1-Traf2-Ask1 induced apoptosis (Nishitoh et al. Genes Dev. 2002 PMID 12050113). In those alternative models, ATF4/CHOP is usually active, but they are not sufficient to cause apoptosis on their own. Unless there is experimental evidence, the authors should not implicate ATF4-CHOP induced cell death in the phenotype.
3. Regarding Wnt/beta-catenin signaling, the authors have now added additional data using Wnt secretion inhibitors to help strengthen the causal relationship between this pathway and Slc7a5 expression, which is an improvement from the previous version of the manuscript.

2nd Revision - authors' response

18 October 2019

Final Response to reviewers

Referee #1

The additional data are convincing and the authors have satisfactorily addressed my questions.

Since the Editor noted that it would not be necessary to provide causality between the induction of

the ISR and the phenotype observed in the *Slc7a5*-deficient mice. In my opinion the manuscript should be published in EMBO Reports.

Referee #2

This manuscript by Poncet and colleagues reports mouse *Slc7a* null embryo phenotypes. Specifically, they describe (1) the *Slc7a* expression pattern during development, (2) report the developmental defects associated with *Slc7a* loss, (3) show that the Integrated Stress Response (ISR) target genes are induced in *Slc7a* null embryos, (4) and show that Wnt/beta catenin signaling regulates *Slc7a* expression. The study is mostly descriptive. The induction of ISR in cells lacking the amino acid transporter *Slc7a* is somewhat predictable. Whether the *Slc7a* loss phenotypes are due to amino acid deprivation, or alternatively due to excessive ISR signaling remains unclear. However, the authors still argue in a few sections of the manuscript that ISR signaling activity is "likely to have a causal role" in the *Slc7a* null phenotype. There is no data supporting such as causal relationship, and therefore, those claims need to be withdrawn. Below are a few specific examples for the authors' consideration.

1. Line 258 - 260: The authors write that "induction of the ISR is the likely underlying cause of these early neural developmental and limb defects." The authors echo such a statement in the Discussion, where the authors write that "these data strongly suggest that rapid induction of the ISR and apoptosis underlie the developmental defects by *Slc7a5* loss." These are overstatements, as there is no evidence presented in this manuscript to establish causality of ISR in mediating the phenotype. On the one hand, ISR activation (GCN2 and eIF2a phosphorylation) as a mere consequence of amino acid transport defect is an outcome that is predictable based on the well-established literature. But the authors need to consider the very likely scenario that the developmental phenotypes are due to specific tissues being deprived of essential amino acids for their function, and not having anything to do with excessive ISR signaling itself. If the authors wish to implicate ISR signaling, they need to repeat the experiment in mice with impaired ISR signaling and demonstrate a rescue in phenotype.

We have moderated these statements and tracked changes. We now make clearer still that we have identified the ISR as a potential pathway upstream of apoptosis induction in *Slc7a5*-null embryos.

2. Line 315: After showing that there is enhanced cell death in *Slc7a5* null embryos, the authors conclude that the results are "consistent with the local induction of the ISR following *Slc7a* loss and the triggering of apoptosis as stress levels rise on a cell by cell basis." I suggest taking out the inference of ISR in apoptosis induction here, as there could be many other causes of cell death. We have removed the inference to ISR here.

The authors should be mindful that there are many other models of UPR stress-induced cell death that does not involve ATF4/CHOP, which includes the role of IRE1-mediated caspase activation (Upton et al., Science 2012 PMID 23042294), Ca/calmodulin-induced apoptosis (Timmins et al., J. Clin. Invest. 2009 PMID 19741297), and IRE1-Traf2-Ask1 induced apoptosis (Nishitoh et al. Genes Dev. 2002 PMID 12050113). In those alternative models, ATF4/CHOP is usually active, but they are not sufficient to cause apoptosis on their own. Unless there is experimental evidence, the authors should not implicate ATF4-CHOP induced cell death in the phenotype.

We do not find well known transcriptional targets of these other cell death pathways (which appear so far described only as induced following ER stress) these include ATF6 pathway targets BiP, GRP94, PDI and IRE-1 pathway targets EDEM1, ERDJ4, ERDJ6, in our E8.5 RNAseq data. In contrast, *Trib3* and *Chac1* are highly expressed. This suggests that these alternative cell death mediators are not activated early in response to *Slc7a5* loss. However, we can not rule out involvement at later stages. This is now noted in the Discussion Lines 429-31.

3. Regarding Wnt/beta-catenin signaling, the authors have now added additional data using Wnt secretion inhibitors to help strengthen the causal relationship between this pathway and *Slc7a5* expression, which is an improvement from the previous version of the manuscript.

Corresponding Author Name: Kate Storey

Journal Submitted to: EMBOR

Manuscript Number: EMBOR-2019-48469